# Catalytic Liquefaction of Kraft Lignin with Solvothermal Approach

**Woldemichael Sebhat** [1], **Ayman El Roz** [1], **Pascal Fongarland** [2], **Léa Vilcocq** [2,*] and **Laurent Djakovitch** [1,*]

1. CNRS, IRCELYON, Université de Lyon, Université Claude Bernard Lyon 1, UMR 5256, 2 Avenue Albert Einstein, CEDEX, F-69626 Villeurbanne, France; wmichael.sebhat@gmail.com (W.S.); ayman.roz@hotmail.com (A.E.R.)
2. CNRS, CPE-Lyon, Catalysis, Polymerisation, Processes, Materials (CP2M), Université de Lyon, Université Claude Bernard Lyon 1, UMR 5128, 2 Avenue Albert Einstein, F-69616 Villeurbanne, France; pfo@lgpc.cpe.fr
* Correspondence: lvi@lgpc.cpe.fr (L.V.); laurent.djakovitch@ircelyon.univ-lyon1.fr (L.D.); Tel.: +33-4-72-43-17-61 (L.V.)

**Abstract:** Lignin is a natural biopolymer present in lignocellulosic biomass. During paper pulp production with the Kraft process, it is solubilized and degraded in Kraft lignin and then burned to recover energy. In this paper, the solvolysis of Kraft lignin was studied in water and in water/alcohol mixtures to produce oligomers and monomers of interest, at mild temperatures (200–275 °C) under inert atmosphere. It was found that the presence of alcohol and the type of alcohol (methanol, ethanol, isopropanol) greatly influenced the amount of oligomers and monomers formed from lignin, reaching a maximum of 48 mg·$g_{lignin}^{-1}$ of monomers with isopropanol as a co-solvent. The impact of the addition of various solid catalysts composed of a metal phase (Pd, Pt or Ru) supported on an oxide ($Al_2O_3$, $TiO_2$, $ZrO_2$) was investigated. In water, the yield in monomers was enhanced by the presence of a catalyst and particularly by Pd/$ZrO_2$. However, with an alcoholic co-solvent, the catalyst only enhanced the formation of oligomers. Detailed characterizations of the products with FTIR, $^{31}$P-NMR, $^1$H-NMR and HSQC NMR were performed to elucidate the chemical transformations occurring during solvolysis. The nature of the active catalytic specie was also investigated by testing homogeneous palladium catalysts.

**Keywords:** Kraft lignin; palladium; zirconium oxide; hydrothermal liquefaction; solvothermal liquefaction; vanillin; guaiacol; alkylguaiacols

## 1. Introduction

Lignocellulose is one of the most abundant sources of renewable carbon on earth. It has been identified as a renewable source of energy and chemicals, and is therefore used as is in various applications from energy to animal feed, being as well an ideal candidate for replacing some of our chemical needs from the petroleum chain. In this approach, lignocellulose appears to be vital for the production of chemicals with a reduced carbon footprint compared to fossil resources. It is composed of three main biopolymers: cellulose (30–50%) and hemicelluloses (15–35%), mainly composed of sugars moieties, and lignin (10–30%), a polyaromatic material [1].

In paper and pulp industry, lignocellulose is fractioned to produce chemical paper pulp from cellulosic fraction. If cellulose is well valorized through such pulping processes, valorization of the two other components (i.e., lignin and hemicellulose) is generally limited to low value-added application or they are burned out in most units to sustain the energy needs of pulp process [2]. The most common pulping process is the Kraft process, in which wood chips are cooked with white liquor containing $Na_2S$ and NaOH [3]. During cooking, hemicelluloses and lignin are dissolved and degraded in liquid phase, called black liquor, as sugars, organic acids and Kraft lignin. This black liquor is concentrated and burnt to

recover inorganic matter and generate energy [3]. Nevertheless, for improving the viability and profitability of biorefineries, including paper industries, efficient valorization of lignocellulose must consider every product stream [4,5]. For integrated forest biorefineries (IFBR) development, efficient valorization of lignin that is mainly composed of aromatics is one of the hurdles preventing implementation of an integrated process.

Lignin is composed of three main phenolic monomers: guaiacyl, syringyl, hydroxyphenyl, bearing aliphatic side-chains (functionalized with alcohol or alkene groups) and linked with a variety of inter-unit bonds, which can be ether bonds (e.g., β-O-4), or C–C bonds (e.g., β-β, β-5, 5-5′) (Scheme 1) [6]. During the Kraft process, lignin is extracted from the lignocellulose matrix and partially depolymerized and degraded. α-aryl ether and β-aryl ether bond are cleaved preferentially during Kraft cooking, forming new moieties such as quinone methides. Condensation reactions forming C–C bonds between phenolic units also occur [7–9]. Kraft lignin can be used in materials applications or transformed in various aromatic or aliphatic building blocks through thermochemical, chemical and biochemical pathways [10–12].

**Scheme 1.** Structure of lignin polymer. In blue, the main linkages between phenylpropane units.

Chemical conversion of lignin was investigated in various reaction conditions [13]. In aqueous media, lignin is soluble in basic conditions. Therefore, working under basic conditions where lignin is soluble is preferred for both facilitating handling and improving lignin conversion. The use of catalysts, generally metals supported on metal oxides, was reported as a way to improve conversion of lignin under reductive conditions or oxidative conditions [14,15]. In the absence of oxidant or reductant, the main effect is the depolymerization of lignin in soluble oligomers and the formation of monomers (methoxyphenols) in low yield (inferior to 15%wt) [13]. As an example, aqueous phase reforming (APR) approach was studied by Weckhuysen et al., with different types of lignin [16]: it consists of generating hydrogen in situ to assist the catalytic reductive depolymerization of lignin. Solubilization in neutral conditions and light alkanes production, together with monomeric fragments, were observed. Yield of monomeric products was between 10–15% depending on the lignin. The recalcitrance of lignin to conventional treatments was widely investigated [17] despite numerous reports on lignin depolymerization to produce chemicals [15]. Ether C–O bonds (β-O-4 and α-O-4) present in lignin can be readily cleaved under hydrothermal treatment, while C–C condensed linkages (mainly, 4-O-5, 5-5, β-5, β-β) require harsher conditions [18].

Lignin solvolysis was also studied in various solvents: alcohols, decalin, tetrahydrofuran, dioxane, formic or acetic acids, etc., pure or mixed with basic aqueous solution [19,20]. The organic solvent can enhance the solubility of lignin [21] and/or play the role of hy-

drogen donor [22]. During solvolysis, fragmentation can be followed by condensation reactions compensating depolymerization and taking place though very reactive quinone methide type compounds. Alternatively, release of formaldehyde through elimination of terminal hydroxymethyl group was suggested as precursor for condensation of phenolic structures [12,13]. To minimize such undesired reaction, the use of ethanol was proven to be effective [14] as alcohol act as hydrogen transfer agents stabilizing reactive phenolics as well as helping to depolymerization through hydrogenolysis [14–16]. Solvolysis results in partial or complete solubilization of Kraft lignin (e.g., Kraft lignin is 90% soluble at 200 °C in basic aqueous solution, and 33% soluble at 200 °C in methanol) [23]. In the absence of catalyst, the production of monomers during solvolysis is generally inferior to 4%wt for Kraft lignin.

Therefore, a heterogeneous catalyst is often used in lignin solvolysis. For example, Kraft lignin was depolymerized in various solvents under inert atmosphere using $Pt/Al_2O_3$ [24], $Mo_xC/C$ [25–27], $Mo_2N/Al_2O_3$ [28], Pd/C [29], Cu-Mo/ZSM5 zeolite [30] and $Ni-Re/Nb_2O_5$ [31]. The yields in monomers depend on different reaction parameters: temperature, solvent, reaction time, catalyst loading. As a general trend, yields superior to 50%wt are only obtained at high temperature (280 °C and higher). Despite numerous works on heterogeneous catalysis for lignin solvolysis, there is a lack of comparative studies of metals and oxide supports used as catalysts for technical lignin solvothermal liquefaction.

The aim of this work is the investigation of Kraft lignin solvolysis in water and organic solvents in the absence or presence of metal supported catalysts. The roles of solvent, metal and support in lignin solubilization and depolymerization are detailed.

## 2. Results and Discussion

### 2.1. Characterization of Kraft Lignin

Kraft lignin was purchased from Sigma Aldrich (St. Louis, MO, USA). Its chemical composition is summarized in Table 1. It corresponds to the composition of a technical lignin: the content in sulfur and inorganics is high and lignin is present as soluble lignin and insoluble (Klason) lignin.

**Table 1.** Chemical composition of Kraft lignin.

| Water Content (%wt) | 6 |
|---|---|
| Acid-soluble lignin (%wt) | 39.5 |
| Klason lignin (%wt) | 47.3 |
| Sugar content (%wt) | 1.4 |
| Ash content (%wt) | 20.0 |
| Organic matter composition (%wt) | C 48%; H 5%; O 34%; S 4% |
| Inorganic matter (ashes) composition (%wt, lignin basis) | Al 1.2%; K 1.2%; Na 1.2%; Mg 720 ppm; Ca 600 ppm; Fe 240 ppm; Li 240 ppm; V 50 ppm. |

### 2.2. Hydrothermal Approach

First, Kraft lignin solvolysis in pure water was investigated (i.e., hydrolysis). The studied Kraft lignin was highly soluble in water without adding any additive. The pH of a $10 \text{ g·L}^{-1}$ solution is around 9 without adding any base.

#### 2.2.1. Non-Catalytic Studies

In a first approach, lignin was solubilized in water and hydrolysis was performed under inert atmosphere without any catalyst.

The effect of the reaction time during hydrothermal treatment at 225 °C and 40 bar was investigated. First, the initial lignin solution was fractionated for comparative purposes. Around half of the lignin precipitated under acidic conditions (Klason phase, KP), 46%wt remained soluble in water (Aqueous phase, AP) and 2%wt was extracted with dichloromethane (Organic phase, OP). Samples treated at different run duration were fractionated; the results are summarized on Figure 1. The Klason phase increased slightly

after 1 h of reaction compared to the untreated sample, but then decreased with reaction time. The aqueous phase also decreased with reaction time. The organic phase increased with reaction time, and doubled after 24 h. The mass balance also decreases with time, it was noticed with longer treatment some solid deposition on the reactor walls hard to retrieve and quantify. Hence, the slight decrease of the Klason phase is probably due to the formation of char-like material, lost inside the reactor. On the other hand, during the fractionation step, the different samples were dried under reduced pressure to remove the water used as a solvent, consequently a part of volatile components formed from the lignin are potentially lost during this step. Estimations measured by evaporating and drying a sample of the reaction mixture under reduced pressure revealed that between 8% at 1 h reaction to 15% at 24 h is lost during such reaction mixture treatment. Thus, material loss is more probably due to combination of both char formation inside the reactor and evolution of volatile compounds in gas phase during reactor depressurization and reaction mixture treatment.

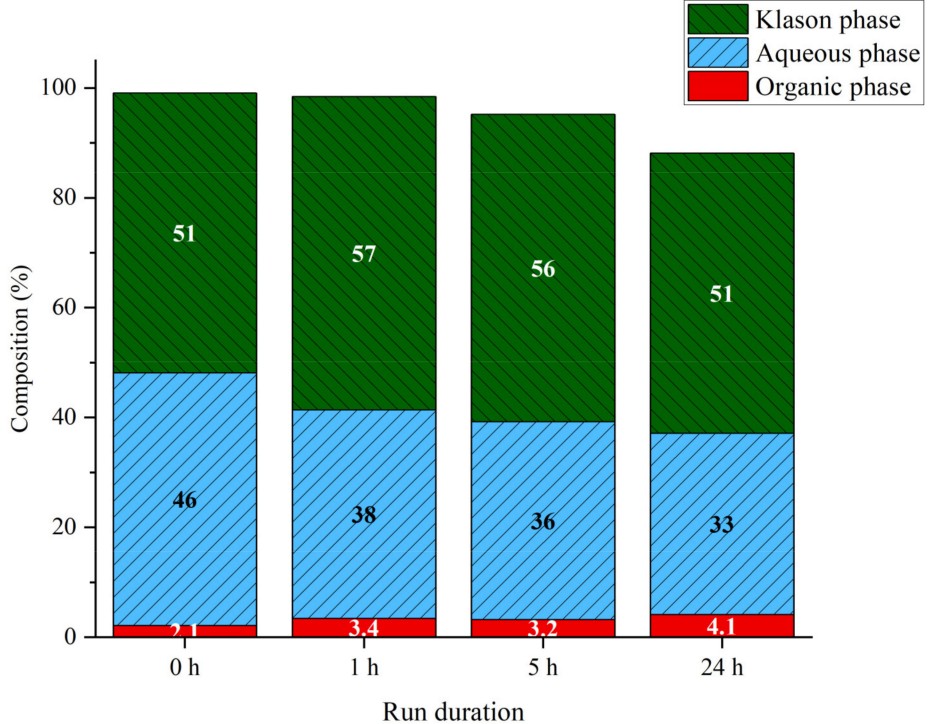

**Figure 1.** Phase compositions of lignin aqueous solutions at different run durations. Reaction conditions: Kraft lignin 10 g·L$^{-1}$, 225 °C, 40 bar Ar.

The GC analysis of the organic phase shows the formation of small concentrations of phenolic monomers: vanillin, guaiacol, guaiacylacetone, acetovanillone, dimethoxyphenol. These products are not present in the initial lignin; they appeared after 1 h of hydrothermal treatment. However, longer treatment does not necessarily produce higher amounts of monomeric products (Figure 2). Guaiacol is the only product that was affected by reaction time; its yield reached a maximum of 8 mg·g$_{lignin}^{-1}$ after a treatment of 24 h. Guaiacol can be assumed to be a stable end product in this case. The total of monomers increased drastically at long reaction times and reached 14.6 mg·g$_{lignin}^{-1}$ after 24 h. In the literature, Kraft lignin hydrolysis in basic solution in medium basic solution (pH = 10) yielded 1.1%wt monomers at 200 °C in 8 h, in accordance with our results. Higher pH gave higher monomers yields, up to 13%wt [23].

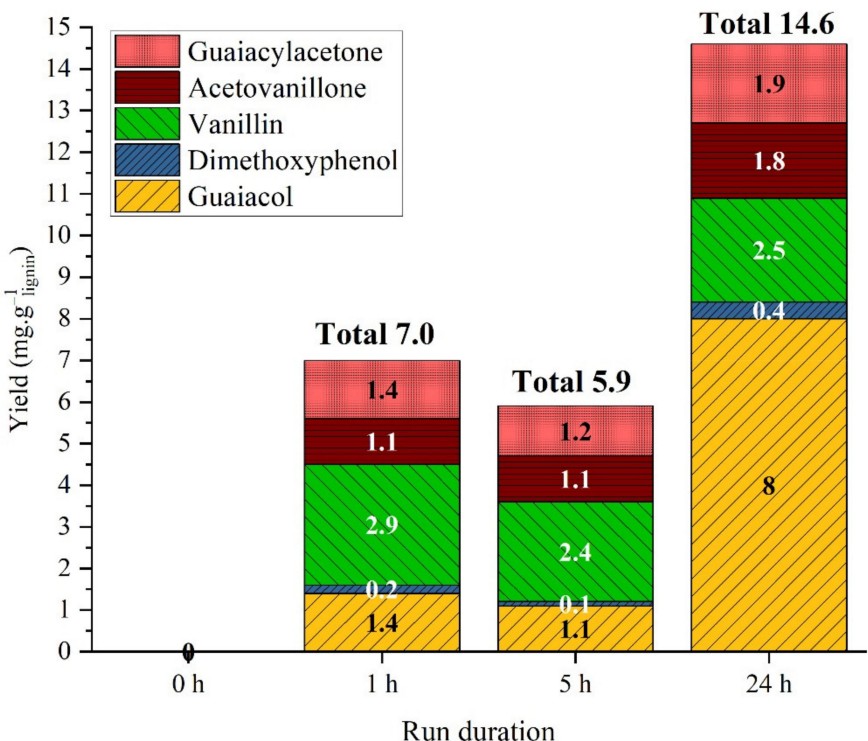

**Figure 2.** Yield of monomeric products in the organic phase at different run durations. Reaction conditions: Kraft lignin 10 g·L$^{-1}$, 225 °C, 40 bar Ar.

Noticeably, for reaction time above 7 h, new products were detectable in traces, namely phenol and p-ethylguaiacol. For a 7 h treatment, the yields were below 0.2 mg·g$_{lignin}$$^{-1}$. After 24 h treatment, the yield of phenol was 0.2 mg·g$_{lignin}$$^{-1}$ and that of p-ethylguaiacol 0.5 mg·g$_{lignin}$$^{-1}$.

The effect of temperature was studied at a relatively high reaction time of 7 h and 40 bar argon, except for the run at 275 °C that was conducted at a pressure of 60 bar to guarantee that aqueous solution remained liquid; it was assumed that the pressure of inert gas has little effect compared to the temperature. The temperature had an important impact on the products distribution (Figure 3). The results show that elevated temperatures resulted in the formation of insoluble products (char) through repolymerization and other side reactions, mainly from the Klason phase. At 225 °C, there was no insoluble fraction at the end of the treatment. The effect of temperature is more pronounced on the Klason phase as it decreased by an order of magnitude of seven, going from 55%wt to 8%wt at 275 °C. Elevated temperature also resulted in poor mass balance that can be linked to important amount of unrecovered material observed on the reactor walls and to the formation of gases ($CO_2$, $H_2$, light alkanes and alcohols, not quantified in this study). The organic phase increased slightly with temperature, maybe through the degradation of aqueous phase.

The main outcome concerning the increase in temperature is the increased organic phase from 3%wt to 7%wt at 225 °C and 275 °C, respectively. Additionally, some major differences were observed in the composition of the organic phase (Table 2). At elevated temperatures, catechols were detected whereas they were not observed at lower temperature. Thus, the data presented in Table 2 show that the yield of aromatic compounds increased with increasing the temperature; at 250 °C, very high amount of guaiacol was observed while other products did not increase in the same manner. Increasing the temperature thus favored catechols derivatives versus guaiacols. The degradation of guaiacol into catechol at high temperature was already reported [32]. Even though at 250 °C higher amounts of aromatics were obtained, the formation of insoluble fractions can be problematic. Therefore, further experiments were examined only at 225 °C.

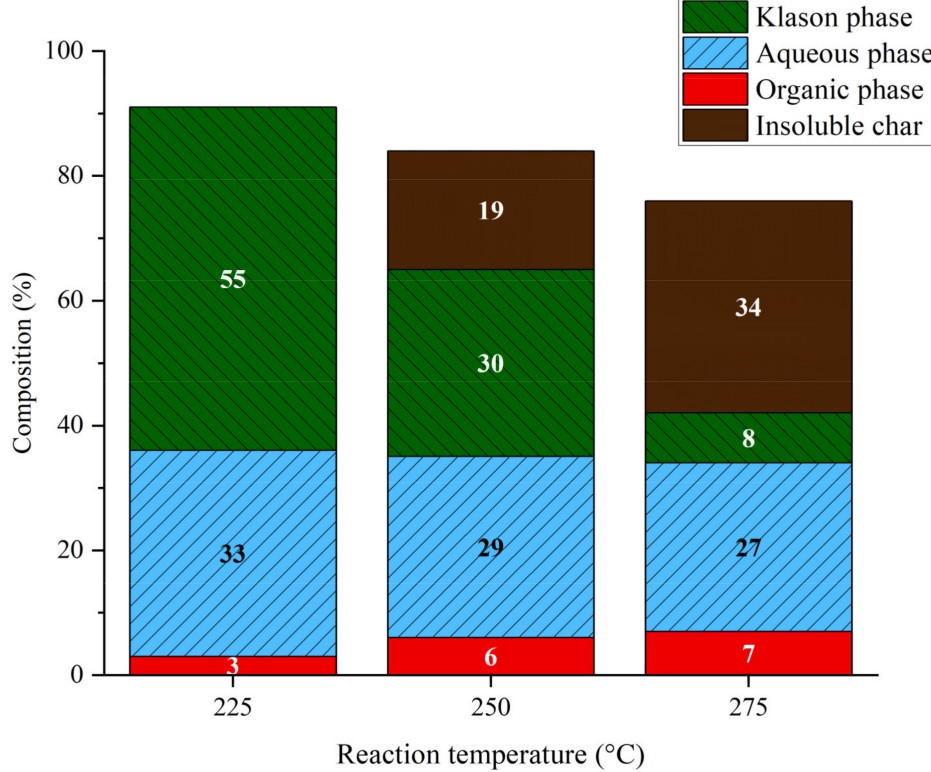

**Figure 3.** Phase compositions of lignin aqueous solutions at different temperatures. Reaction conditions: Kraft lignin 10 g·L$^{-1}$, 7 h, 40 bar Ar (225 and 250 °C) or 60 bar Ar (275 °C).

**Table 2.** Yield of monomeric products in the organic phase at different temperatures.

| Monomeric Products | Yield (mg·g$_{lignin}$$^{-1}$) | | |
|---|---|---|---|
| | **225 °C** | **250 °C** | **275 °C** |
| Phenol | - | 0.5 | 0.4 |
| *p*-Cresol | - | - | 0.1 |
| Catechol | - | - | 3.6 |
| 4-Methylcatechol | - | - | 2.2 |
| Guaiacol | 4.4 | 15.2 | 1.2 |
| 3-Methylcatechol | - | - | 0.2 |
| m-Creosol | - | 0.4 | - |
| *p*-Creosol | - | 1.1 | 0.8 |
| 2,6-Dimethyl-hydroquinone | - | - | 0.1 |
| 4-Ethylcatechol | - | - | 0.2 |
| *p*-Ethylguaiacol | - | 1.1 | 1.0 |
| Vanillin | 2.5 | 2.7 | 0.2 |
| **Total identified** | **10.1** | **24.4** | **10.5** |

Reaction conditions: Kraft lignin 10 g·L$^{-1}$, 7 h, 40 bar Ar (225 and 250 °C) or 60 bar Ar (275 °C).

Finally, hydrothermal approach of Kraft lignin solvolysis led to a partial depolymerization of soluble lignin and Klason lignin at 225 °C. At higher temperature, solubilization of lignin increased drastically. Reaction time was not a critical parameter. Monomers, and particularly guaiacol, were produced in small quantities at 225 °C. Monomers yields increased with reaction time and reaction temperature.

### 2.2.2. Catalytic Studies

In order to increase monomers yield, a heterogeneous catalyst was added to the reaction medium during hydrothermal solvolysis of Kraft lignin.

Three different metals, platinum, palladium and ruthenium, were used for catalytic solvolysis of Kraft lignin. They were chosen for their hydrogenation and hydrogenolysis activities. Metal nanoparticles were dispersed on three different supports: titanium dioxide, alumina and zirconium dioxide. Metal loading varied from 1.4 to 2.4%wt (Table 3).

**Table 3.** Physical and chemical properties of catalysts.

| Catalyst | Metal Loading (%wt) | Mole of Metal per Gram [a] ($\mu$mol) | Specific Surface Area BET [b] ($m^2 \cdot g^{-1}$) |
|---|---|---|---|
| Pt/TiO$_2$ | 1.4 | 7.2 | 89 |
| Pt/Al$_2$O$_3$ | 2.3 | 7.2 | 302 |
| Pt/ZrO$_2$ | 2.4 | 8.3 | 58 |
| Pd/ZrO$_2$ | 2.0 | 8.5 | 61 |
| Ru/ZrO$_2$ | 1.7 | 7.0 | 57 |

[a] From ICP measurements. [b] From N$_2$ physisorption isotherms.

Prepared catalysts were evaluated in a batch reactor under reference conditions: 225 °C, 40 bar of argon, 1000 rpm for 3 h. The solution contained 1%wt alkaline lignin, metal to lignin mass ratio was 1%wt for Pt, 0.6%wt for Pd and 0.5%wt for Ru catalysts, keeping the initial molar loading based on Pt catalyst to 76.8 $\mu$mol. When another metal was tested the molar equivalent was added, that is 76.8 $\mu$mol of metal for each run (ca. 10 g of catalyst).

Three inorganic oxides, namely Al$_2$O$_3$, TiO$_2$, and ZrO$_2$ were evaluated as a support for Pt particles. Looking at the phases distribution after Kraft lignin hydrothermal solvolysis (Figure 4), the use of various supports did not generate different results as regards to the organic phase. In the case of alumina, a slight decrease in organic phase is observed. In general, when the catalyst was used the mass balance also decreased slightly, lignin adsorption on the catalyst surface and/or gas production with platinum catalyst might explain this observation [16].

The analysis of the organic fraction shows that the only product affected by the catalyst was guaiacol, those yield increased from 2 mg·g$_{lignin}^{-1}$ without catalyst to 2.8, 5.3 and 3.5 mg·g$_{lignin}^{-1}$ in the presence of Pt/Al$_2$O$_3$, Pt/ZrO$_2$ and Pt/TiO$_2$, respectively (Figure 5). The difference observed could be related to Pt dispersion on support, or to support effects. Indeed, the porosity of support can enhance or hinder the conversion or lignin or oligomeric lignin fragments on the catalytic sites. However, all three supports are mesoporous materials with BET specific surface areas in the same order of magnitude. The acido-basic properties of the supports [33] can also play an important role in lignin conversion. All three supports could bear acid and basic sites, which could play a role in monomers production.

In order to gain insight on the effect of other metals on the conversion of lignin, different catalysts were prepared following a deposition–precipitation procedure using ZrO$_2$ as support. Ruthenium [34] and palladium [35] based catalysts were described in literature as efficient catalysts for lignin depolymerization. In this work, they were prepared and compared to platinum one. The results (Figure 6) showed that the different catalyst does not create a disruptive difference on phase distribution. A slight difference is observed with the Pd/ZrO$_2$ catalyst, which gave a slightly higher amount of organic phase and aqueous phase compared to the other catalysts. The mass balance was slightly lower when catalysts were used, particularly for Pt/ZrO$_2$ and Ru/ZrO$_2$, which could indicate the formation of gaseous or volatiles compounds or of char-like material (see above). However, these missing materials still represent less than 15% of total mass balance.

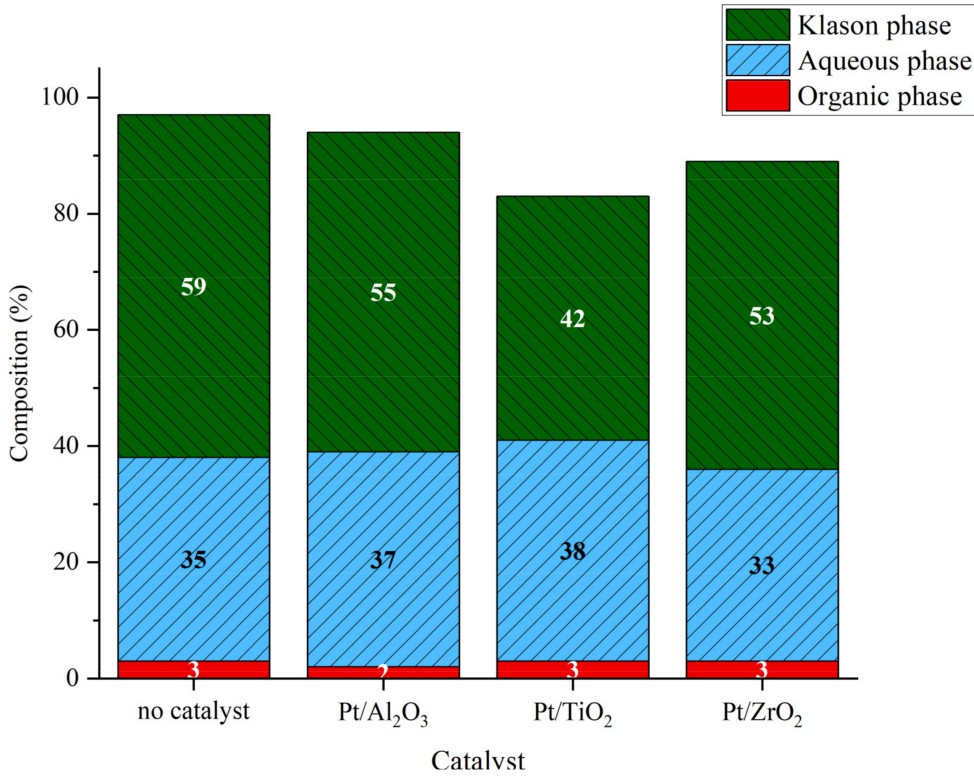

**Figure 4.** Phases composition after lignin catalytic liquefaction over platinum catalysts. Reaction conditions: Kraft lignin 10 g·L$^{-1}$, mass ratio Pt/lignin 1/100, 3 h, 225 °C, 40 bar Ar.

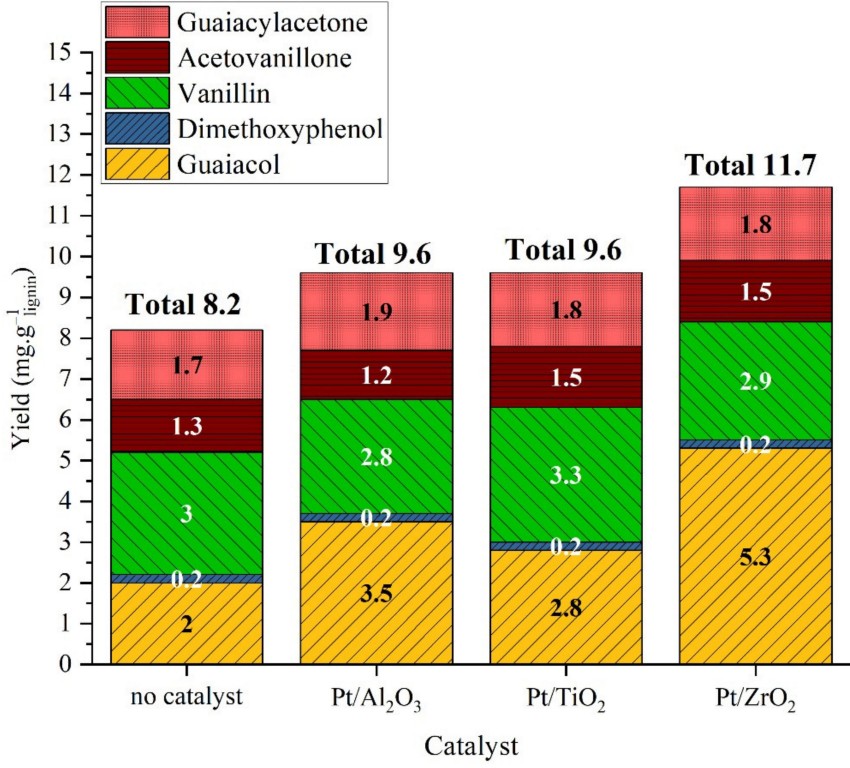

**Figure 5.** Yield of monomeric products in organic phase after lignin catalytic liquefaction over platinum catalysts. Reaction conditions: Kraft lignin 10 g·L$^{-1}$, 3 h, 225 °C, 40 bar Ar.

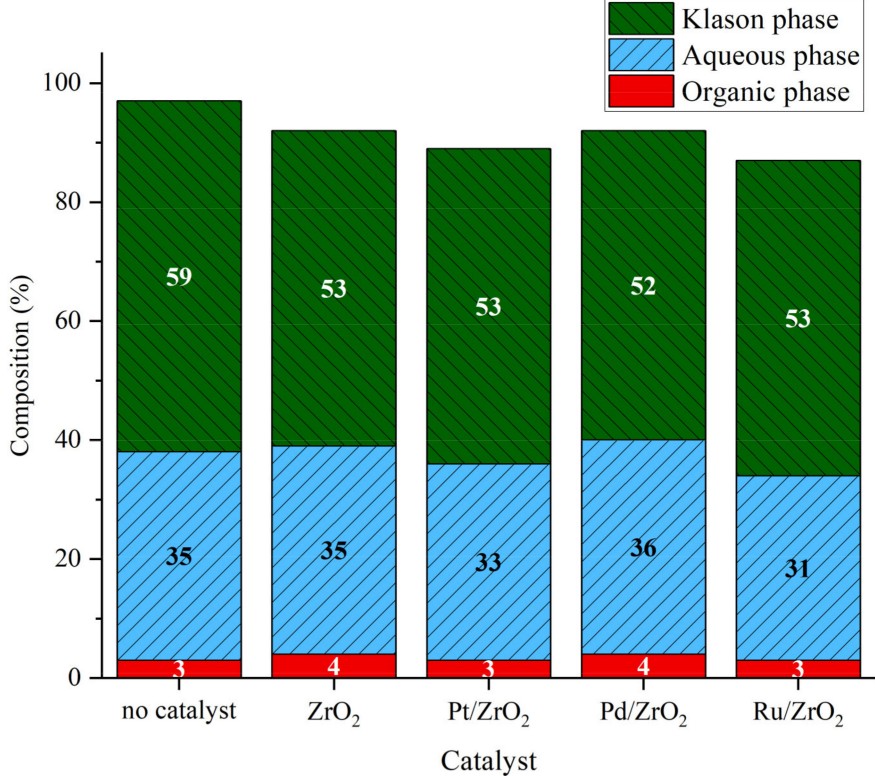

**Figure 6.** Phases composition after lignin catalytic liquefaction over metal catalysts. Reaction conditions: Kraft lignin 10 g·L$^{-1}$, mass ratio Pt/lignin 1/100 or molar equivalent, 3 h, 225 °C, 40 bar Ar.

The analysis of the OP fraction (Figure 7) shows that the different metals have a certain effect on aromatic yields. ZrO$_2$ support alone led to a slight increase in guaiacol yield (from 2 mg·g$_{lignin}$$^{-1}$ to 2.8 mg·g$_{lignin}$$^{-1}$) and in vanillin yield (from 3 mg·g$_{lignin}$$^{-1}$ to 3.7 mg·g$_{lignin}$$^{-1}$). Such a catalytic effect of ZrO$_2$ was observed earlier for Kraft lignin hydrolysis at high temperature [36]. Pt/ZrO$_2$ led to double guaiacol yield but also to decrease vanillin yield, indicating the occurrence of degradation reaction consuming vanillin. Pd/ZrO$_2$ led to increase the yield of guaiacol by a factor 3 while maintaining vanillin yield. Ru/ZrO$_2$ exhibited an intermediate behavior with an increased guaiacol yield by a factor 2 and a steady vanillin yield. Additionally, the reaction pathways for obtaining the different compounds seem to be independent, as the increased yield of guaiacol did not affect that of other products. Therefore, guaiacol is not a degradation product of vanillin, for example. In the literature, different monomers are obtained from lignin in oxidative conditions (vanillin, acetovanillone, vanillic acid mainly) and in neutral to reductive conditions (guaiacol and alkyl guaiacols, or catechol and derivatives at high temperature) [13]. Here, the products obtained in the absence of catalyst are mainly oxidation products (vanillin and acetovanillone) whereas in the presence of a catalyst, and particularly Pd/ZrO$_2$, hydrogenolysis products are in majority (guaiacol), with only a small increase in oxidation products. Therefore, metal catalyst seems to favor the cleavage of inter-unit linkage following a neutral or reductive pathway.

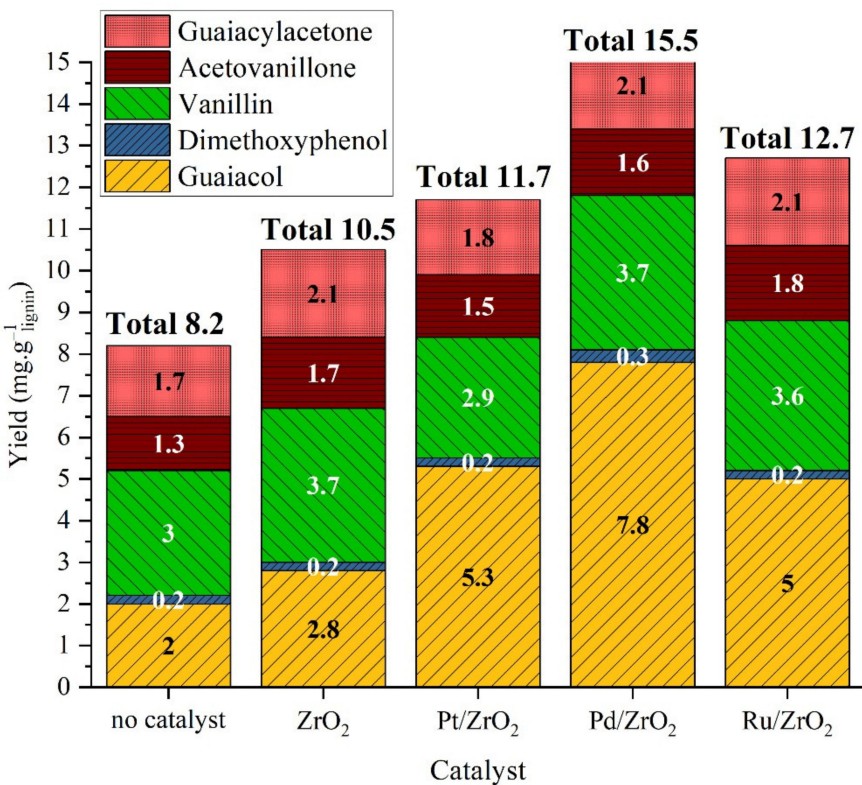

**Figure 7.** Yield of monomeric products in organic phase after lignin catalytic liquefaction over metal catalysts. Reaction conditions: Kraft lignin 10 g·L$^{-1}$, mass ratio Pt/lignin 1/100 or molar equivalent, 3 h, 225 °C, 40 bar Ar.

Finally, adding a catalytic system to hydrothermal solvolysis of Kraft lignin had no effect on lignin solubilization. Therefore, conversion of Klason lignin into soluble lignin and organic products depends mainly on temperature and reaction conditions; the interaction of lignin with heterogeneous catalyst seems limited. However, the monomers' yield sharply increased in the presence of a heterogeneous catalyst. Pd/ZrO$_2$ support was the best compromise between catalytic production of guaiacol and vanillin and limited degradation of monomers such as vanillin. It produced 15.5 mg·g$_{lignin}$$^{-1}$.

### 2.3. Solvothermal Approach

Despite the use of catalysts, when pure water was used lignin conversion and product yields remained limited. Therefore, we evaluated the role of adding alcohol as co-solvent under previously defined conditions, first without adding catalyst.

### 2.3.1. Non-Catalytic Studies

Lignin liquefaction was performed with an alcohol/water mixture at 225 °C. The use of water/alcohol solvent system remarkably alters the phase distribution (Figure 8). The addition of alcohol systematically increased the amount of organic phase and reduced the amount of Klason phase. As the effect of alcohols on the fractionation step is rather limited, the change observed on the phase distribution can be attributed to the participation of alcohols in the depolymerization step rather than solely improved extraction. Ethanol gave the highest amount of organic phase with 45%wt, followed by isopropanol with 38%wt and finally methanol 15%wt. This is a high increment compared to the runs carried out in water (3%wt). The increase in organic phase seems to be to the detriment of the Klason phase. The aqueous phase does not seem to be highly affected by the addition of alcohols except for methanol where an increase of 11% of aqueous phase compared to the run in water alone is observed.

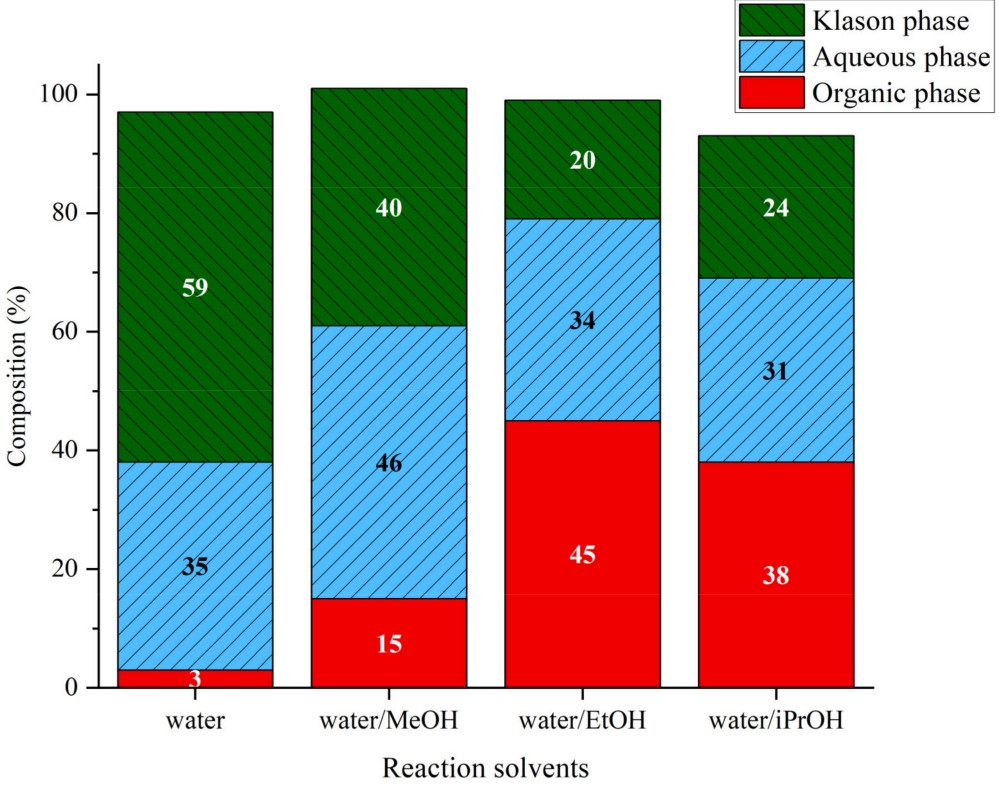

**Figure 8.** Phase compositions after lignin liquefaction in water/alcohol mixtures. Reaction conditions: Kraft lignin 10 g·kg$^{-1}$, 3 h, 225 °C, 40 bar Ar.

The role of alcohol solvents on lignin solvolysis was studied by Kouris et al. [37]. They investigated various alcohols as reaction media for soda lignin solvolysis at mild temperature (200 °C) and observed a correlation between hydrogen bonding ability (related to Hansen solubility parameters (HSP)) and lignin solubilization. In their conditions, methanol was the best solvent for lignin solvolysis. We have applied the same methodology to evaluate the correlation between lignin solubilization represented by HSP for the Kraft lignin used in our conditions, and the level of delignification expressed in function of Klason phase. Hansen parameter approach is based on the determination of solubility parameter δ that is supposed quantifying solute-solvent interaction including dispersion forces ($\delta_D$), molecule polarity ($\delta_R$) and hydrogen bonds ($\delta_H$) according to the theory of regular solutions [38]. HSP approach uses a graphical representation of the mutual solubility by the way of a solubility graph. The more the distance is short between solute and solvent, the more the solubility will increase. Affinity between lignin and solvent is represented by a factor named relative energy difference (RED) according to the following equation: RED = $R_a/R_0$. Where $R_a$ is equivalent to the "distance" between both molecules in the solubility graph and $R_0$ a specific radius of the solute. Parameters of Kraft lignin and the mixture water-alcohol have been taken from literature (ref Hansen) and presented in Table 4.

**Table 4.** Hansen parameters for Kraft lignin solubilization in different solvents mixtures.

|  | $\delta D$ | $\delta P$ | $\delta H$ | $R_0$ |
|---|---|---|---|---|
| Kraft Lignin | 21.7 | 14.2 | 16.9 | 13.5 |
| Water-MeOH (50/50) | 15.3 | 14.15 | 32.3 | - |
| Water-EtOH (50/50) | 15.65 | 12.4 | 30.85 | - |
| Water-iPrOH (50/50) | 15.65 | 11.05 | 28.9 | - |

As shown in Figure 9, it can be found also a correlation between the percentage of delignification and the relative energy difference between Kraft lignin and solvent mixtures in line with previous studies on pure solvent, as mentioned before. However, this is just an attempt to try to explain solvent effect just by "solvation" issues. However, we have to remind that HSP approach is supposed to give indication only on solubilization and not on reactivity without any considerations for temperature or pH effects. We can just imagine that solvation capacity is one of the contributions for lignin depolymerization. In our conditions, the order of reactivity of alcohols was EtOH > iPrOH > MeOH (based on Klason lignin conversion and organic phase production).

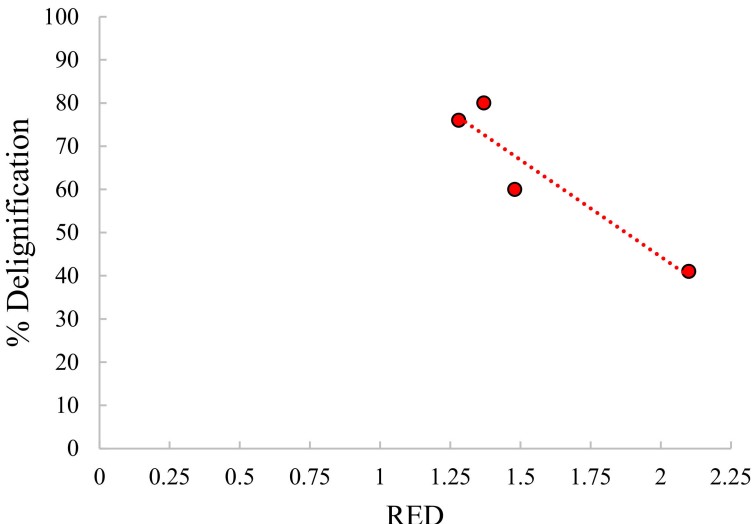

**Figure 9.** Relative Energy Difference (RED) for solvation of Kraft lignin in different solvent mixtures.

Monomeric products present in the organic phase were quantified with GC (Figure 10). The addition of alcohol improved the monomeric yields. The total yield of identified molecules was 8.2, 26.4, 48.2 and 12.0 mg·$g_{lignin}^{-1}$ for water alone, water/ethanol, water/isopropanol and water/methanol, respectively. Yield in guaiacol was particularly enhanced in the presence of alcohols. New products were detected when alcohols were used, mainly guaiacol derivatives with alkyl side chain (para position) resulting from hydrogenolysis of alkyl inter-unit linkages, and from alcohol reaction with aromatics in the reactivity order: isopropanol > ethanol > methanol. It was noted that the double bonds of (iso)eugenol were preserved but their concentrations remain low. Vanillin, one of the major products in water, is highly reduced when alcohols were added. Acetovanillone, which is also a major product in water, was reduced but less affected than vanillin.

Alcohols have a strong impact on monomers production. The presence of alkyl guaiacols when alcohols are used evidences the reaction of alcohols with aromatic moieties of lignin. Propyl-guaiacol is particularly produced in high yield when isopropanol is present. Moreover, alcohols can play the role of hydrogen donors therefore facilitating the cleavage of inter-unit linkages and increasing the monomer content. Iso-propanol is specially known for its ability to give hydrogen (and form acetone) and gives the highest total monomer yield.

### 2.3.2. Catalytic Studies

Following studies in pure water, Pd/$ZrO_2$, which shown to be efficient for the production of guaiacol from lignin, was used in water/alcohol solvents. When Pd/$ZrO_2$ was added the organic phase increased from 4% to 13%, 54%, and 57% for water/MeOH water/EtOH and water/iPrOH, respectively (Figure 11). For water/MeOH, the impact of catalyst on the phase distribution was rather limited and the trend resembled that of water alone. The catalyst improved the liquefaction of lignin to produce dichloromethane-

extractable fragments when ethanol and isopropanol were present. The effect of the catalyst was highly pronounced with isopropanol where the organic phase increased from 38% to 57% and the amount of the Klason phase was also highly reduced from 24% to 12%. In the case of ethanol, the Klason phase was less affected and remained the same (20%) but the aqueous phase was reduced from 34% to 23%. These observations can be related to the capacity of alcohols to act as efficient hydrogen transfer agents in the presence of palladium-based catalysts [39].

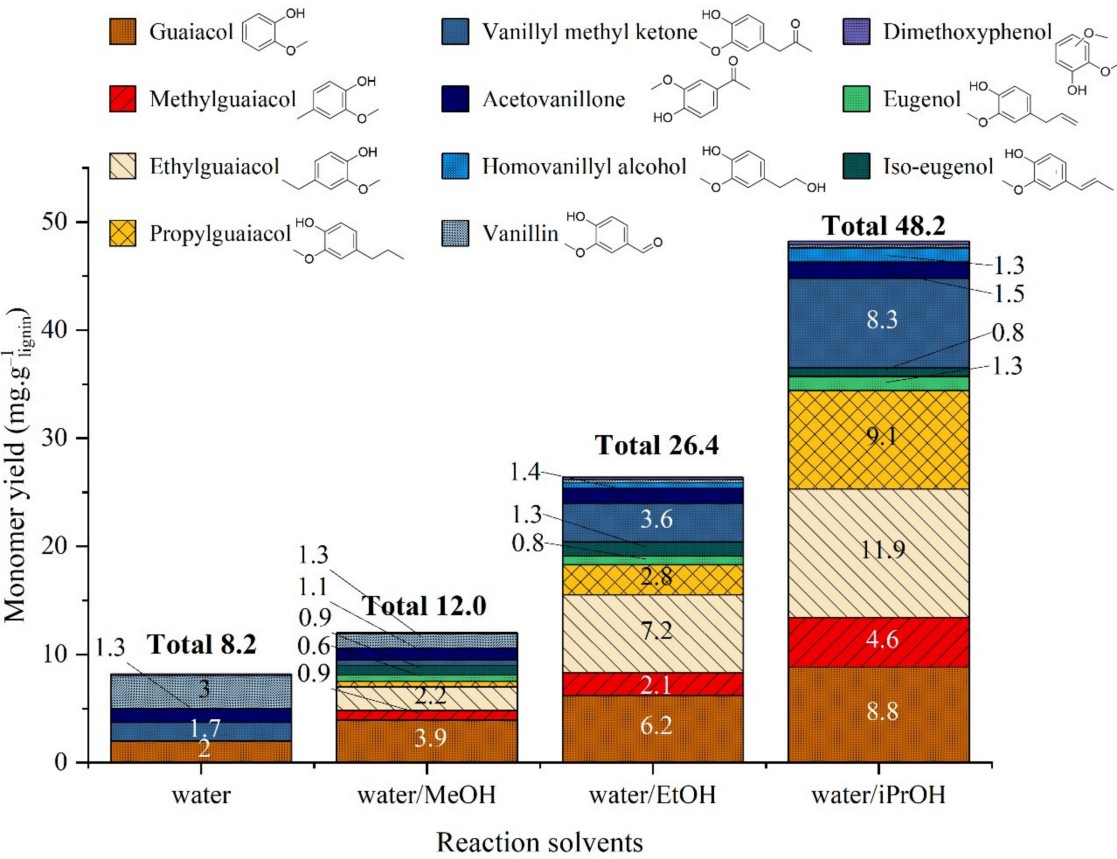

**Figure 10.** Yield of monomeric products in organic phase after lignin liquefaction in water/alcohol mixtures. Reaction conditions: Kraft lignin 10 g·kg$^{-1}$, 3 h, 225 °C, 40 bar Ar.

A run was performed with water/EtOH mixture but argon was replaced by hydrogen. When the reaction was performed in the presence of Pd/ZrO$_2$ and EtOH under hydrogen, the Klason phase was similar to the one obtained without H$_2$, but the yield in aqueous phase was higher (31%). Surprisingly, the presence of hydrogen decreased slightly the amount of organic phase (47%).

The increment observed in the organic phase for ethanol and isopropanol was not necessarily accompanied with an increase of global monomer yields (Figure 12). However, one can note that the distribution of monomers was modified by the presence of the catalyst: less alkylguaiacols were formed but more guaiacol, which may be the marker of hydrogenolysis of alkyl chains in alkylguaiacols in the presence of a heterogeneous catalyst. More acetovanillone and vanillin formed in MeOH/water but less (iso)eugenols.

Even though we cannot propose an interdependence between the products at this stage, the run under hydrogen produced a higher amount of identified aromatics, suggesting that hydrogenolysis and/or hydrogenation reactions are important for the production of monomers. These results are rather close to those observed when performing the reaction in water/iPrOH mixture; iPrOH being known to act as hydrogen donor we can conclude that heterogeneous Pd/support, here ZrO$_2$, acts as a catalyst performing hy-

drogenolysis under the reaction conditions. That is observed for both the reaction involving Pd/ZrO$_2$ in Water/$i$PrOH mixture and the reaction involving Pd/ZrO$_2$ under hydrogen in Water/EtOH mixture.

Finally, the role of catalyst in lignin solvolysis is not to produce more monomers, but rather than to increase the amount of organic phase soluble products. The mechanism of depolymerization is certainly different in the presence of a catalyst, which explains the difference in phase distribution. Pd seems to have a strong impact on the production of oligomers present in the organic phase. The Pd-based catalyst may also have caused coupling and condensation reaction of monomers, thus reducing the number of detectable aromatics.

### 2.3.3. Characterization of Reaction Products

In order to get more insights on the mechanism of solvolysis of Kraft lignin, the organic phase and Klason phase fractions obtained after treatment with water and mixtures with alcohol in the absence of a catalyst were analyzed with FTIR, $^{31}$P- and HSQC-NMR.

The Klason phases from the different treatments were analyzed with FT-IR (Figure 13). In the initial lignin, bands corresponding to aromatic bonds (1602, 1515, 1428, 1140 cm$^{-1}$), aliphatic bonds (1362, 1461 cm$^{-1}$), C–O bond (1086 cm$^{-1}$) and carbonyl (1650 cm$^{-1}$, weak) are observed; particularly, bands at 1268 and 1149 cm$^{-1}$ characteristic of guaiacyl unit are visible. After solvolysis, the main change is the higher intensity and shift of C=O band to higher frequencies (1650 to 1709 cm$^{-1}$), maybe due to removal of the salts initially present in lignin material during the treatment of the reaction mixture [40]. However, we cannot exclude the formation of unconjugated carbonyl groups on residual lignin those vibrations are expected in that region. Some changes also appear in the aromatic bands, with the increased intensity of band at 1515 cm$^{-1}$. Band corresponding to ether or alcohol C–O bonds (1086 cm$^{-1}$) are more visible after solvolysis, as well as band at 1362 cm$^{-1}$ that could correspond to phenol. The spectrum for Klason phase from hydrothermal reaction and reaction with methanol are quite similar and less resolved. In general, the Klason phase preserved most of the functionalities present in the initial lignin, with minor changes indicating the partial oxidation of lignin (more carbonyls and alcohols groups).

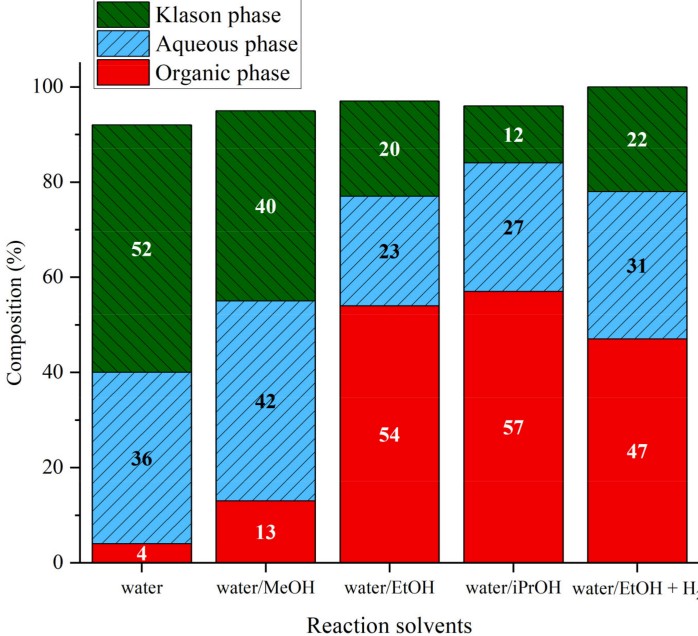

**Figure 11.** Phase compositions after lignin liquefaction in water/alcohol mixtures over Pd/ZrO$_2$. Reaction conditions: Kraft lignin 10 g·kg$^{-1}$, mass ratio Pd/lignin 0.6/100, 3 h, 225 °C, 40 bar Ar.

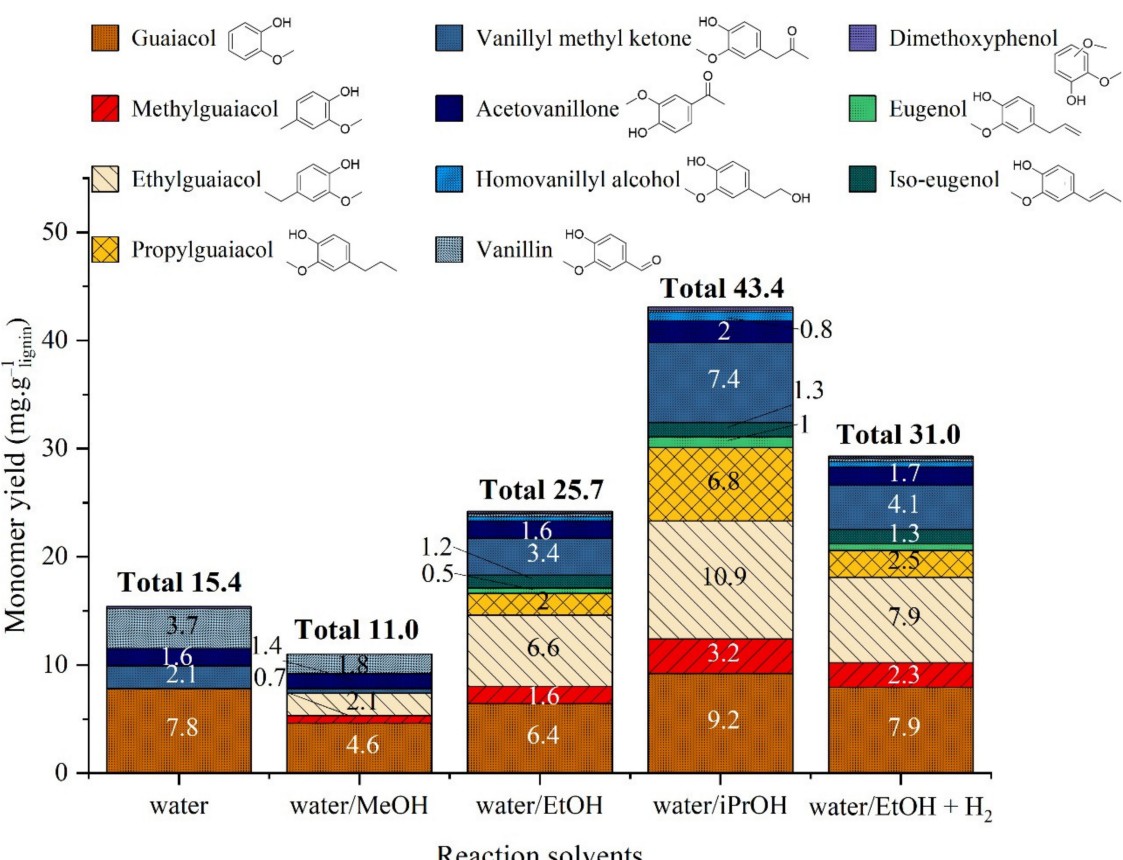

**Figure 12.** Yield of monomeric products in organic phase after catalytic lignin liquefaction in water/alcohol mixtures over Pd/ZrO$_2$. Reaction conditions: Kraft lignin 10 g·kg$^{-1}$, mass ratio Pd/lignin 0.6/100, 3 h, 225 °C, 40 bar Ar.

Free hydroxyl groups can be quantified by $^{31}$P-NMR after phosphitylation with 2-Chloro-4,4,5,5-tetramethyl-1,3,2-dioxaphospholane (TMDP) (Figure 14). The initial lignin is not soluble in organic solvents because it contains salts. To extract the salts, the lignin was precipitated with sulfuric acid and was washed multiple times; the Klason lignin obtained represents only one fraction, so quantitative data representative of the lignin used for experiments cannot be obtained. Instead, data corresponding to Klason lignin in the initial Kraft lignin were presented. A total of 6 mmol·g$^{-1}$ of −OH groups were observed, composed of condensed phenolics, guaiacyl and aliphatic −OH in majority. Hydroxyben-zyl and carboxylic −OH groups were also present in minor amounts. The total hydroxyl groups for samples from treatment in water and water/methanol remained almost the same compared to the initial Klason lignin. For ethanol and isopropanol treated samples, a slight increase in total −OH groups is observed. This can be explained by the C–O–C cleavage making visible oxygen atoms trapped in ether bond, but also by alkylation of alcohols on the lignin. The condensed phenolics increased after solvolysis, and particularly when methanol was used as a co-solvent, suggesting that some recondensation reactions occurred. The amount of aliphatic hydroxyls decreased when water and water/methanol were used as solvents. When ethanol or isopropanol were used as a co-solvent, all types of −OH groups increased, indicating a general oxidation of Klason lignin, in line with the higher reactivity observed below with this mixture of solvents. Oxidation can occur with OH$^-$ coming from basic aqueous medium. In this case, the observed condensed phenolics can be the recalcitrant fraction of lignin that EtOH and iPrOH treatment will not cleave; and after liberation of less recalcitrant fragments these structures were concentrated.

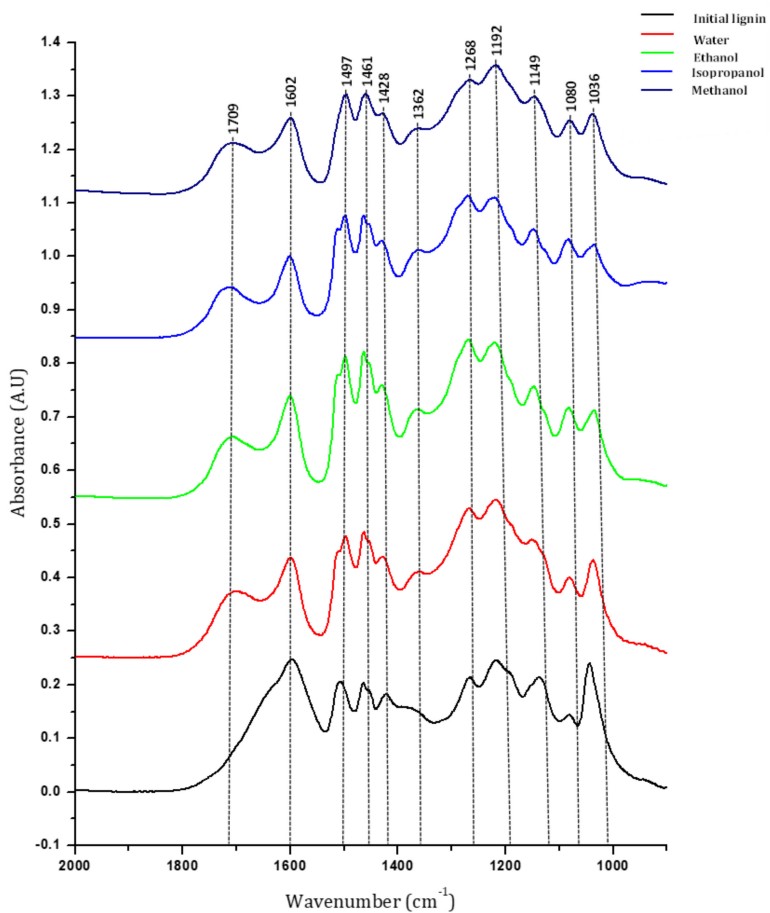

**Figure 13.** FT-IR spectrum of initial lignin and Klason phases after treatment in different water and water/alcohol mixtures.

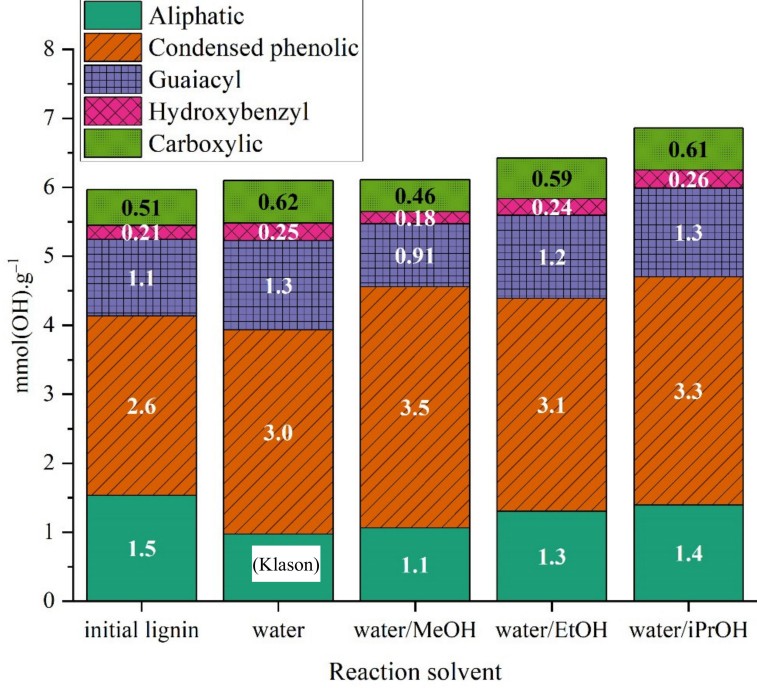

**Figure 14.** Concentration of -OH groups in Klason phases obtained in different solvent mixtures without catalyst.

The analysis of the organic phases with $^{31}$P-NMR showed major differences between the samples (Figure 15). As previously mentioned, the data should be put in perspective with the amount of organic phases isolated, which are 3, 15, 38 and 45%wt for water, water/MeOH, water/EtOH and water/iPrOH, respectively. The total hydroxyl groups were increased in the sample obtained from water and water/MeOH solvolysis reactions but the two samples show major differences in the distribution of hydroxyl groups. The sample treated with methanol shows the highest amount of condensed phenolics followed by isopropanol. For methanol the increase can be due to the formaldehyde formation known to react with alkali lignin with the free 5-positions in the phenolic (guaiacyl) moieties (methoxylation), and the side chain when carbonyl group exists delivering in fine more condensed structures [41]. The case of isopropanol is intriguing knowing that acetone do not act as formaldehyde. The samples treated in water and water/EtOH have slightly lower amounts of condensed phenolics. As guaiacol was the main monomer observed, it was expected to find an increased amount of guaiacyl −OH groups in all samples. Carboxylic groups also increased, particularly for water and water/methanol solvents. Aliphatic alcohols were less present after solvolysis. It should be noted that $^{31}$P-NMR analysis after phosphitylation quantify free hydroxyl groups, hence if O-alkylation occurred with alcohols present in the reaction mixture the result can be biased [42–44].

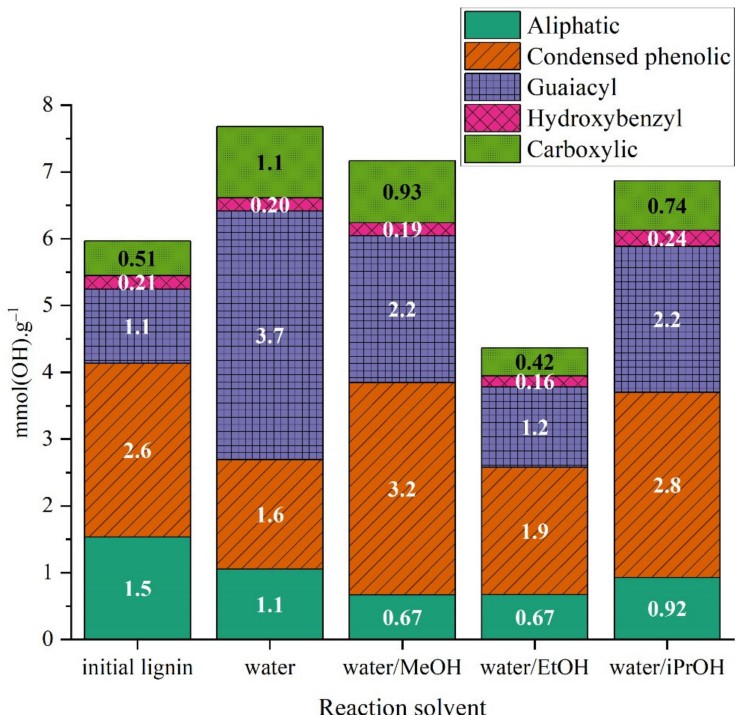

**Figure 15.** Concentration of OH groups in organic phases obtained in different solvent mixtures in the absence of catalyst.

The presence of –OH corresponding to condensed phenolics in large amounts confirm the presence of non-identified condensed products (such as biphenyls or larger oligomers) in the organic phase. These products cannot be analyzed with GC methods but can still be present in the organic phase mass.

Organic phases and precipitates were analyzed first by $^1$H NMR (ESI, Figures S3–S11, and Table 5). Spectra of the OP phases clearly showed an increase of the aliphatic protons in the non-oxygenated region (i.e., 0.5–2.3 ppm), particularly when alcohol is used as co-solvent. Regarding the oxygenated region, generally a decrease is observed, while it remains limited when pure water is used. Regarding the analysis of Klason phase, the general tendency indicates limited decrease of the proton content in aliphatic zones, except

when a mixture of water/MeOH is used that showed an increase of the content associated with non-oxygenated aliphatic proton.

**Table 5.** Semi-quantitative determination of the aliphatic protons occurrence in NMR spectra calculated based on 100 aromatic protons.

| | Initial Lignin | Water | | Water/MeOH | | Water/EtOH | | Water/iPrOH | |
|---|---|---|---|---|---|---|---|---|---|
| | | OP | KP | OP | KP | OP | KP | OP | KP |
| $H_{aliphatic}$ | 69 | 113 | 51 | 316 | 104 | 259 | 91 | 294 | 77 |
| $H_{oxygenated\ aliphatic}$ | 228 | 212 | 133 | 137 | 169 | 144 | 183 | 137 | 183 |
| $H_{aromatic}$ | 100 | 100 | 100 | 100 | 100 | 100 | 100 | 100 | 100 |

In few cases, an increase of the proton associated with carboxylic acids function was observed for OP fraction, particularly in pure water and water/EtOH mixture (i.e., from 7 to ca. 28 $H_{COOH}$/100 $H_{Aromatic}$). The same observation can be made for KP, in a lesser extend (i.e., from 7 to ca. 16 $H_{COOH}$/100 $H_{Aromatic}$) but for all reaction conditions.

The HSQC-NMR (Heteronuclear Single Quantum Coherence—Nuclear Magnetic Resonance spectroscopy) spectra of different organic phases confirm this analysis (ESI, Figures S12–S20, and Figures 16–18—reaction in water/EtOH). Thus, the number of aliphatic proton raised from 34 Aliphatic Units per 100 Aromatic Units in the initial lignin to 120 in Water, 389 in Water/MeOH, 230 Water/EtOH and 368 Water/iPrOH. The same tendency, while less pronounced was observed for the KP phase where Aliphatic units raised to 120 in Water, 117 in Water/MeOH, 75 in Water/EtOH and 60 in Water/iPrOH. Thus, data showed that methylation, ethylation and propylation of the aromatic structures occurred during the reaction. The direct propylation of the aromatic ring seems unlikely in our condition, propylphenyl products detected with GC-MS (see Figure 9) are more likely obtained from the cleavage of β-O-4 linkages, followed by dehydration, hydrogenolysis, and hydrogenation (through hydrogen transfer) reactions. This is supported by the fact that in almost all HSQC spectra the correlation corresponding to the β-O-4 linkages (69.05/5.14 (δC/δH)) decreased strongly (Figure 16), dropping thus from 2.8 β-O-4 linkages per 100 Aromatic Units to 0 in OP phases and 0–0.4 in KP phases.

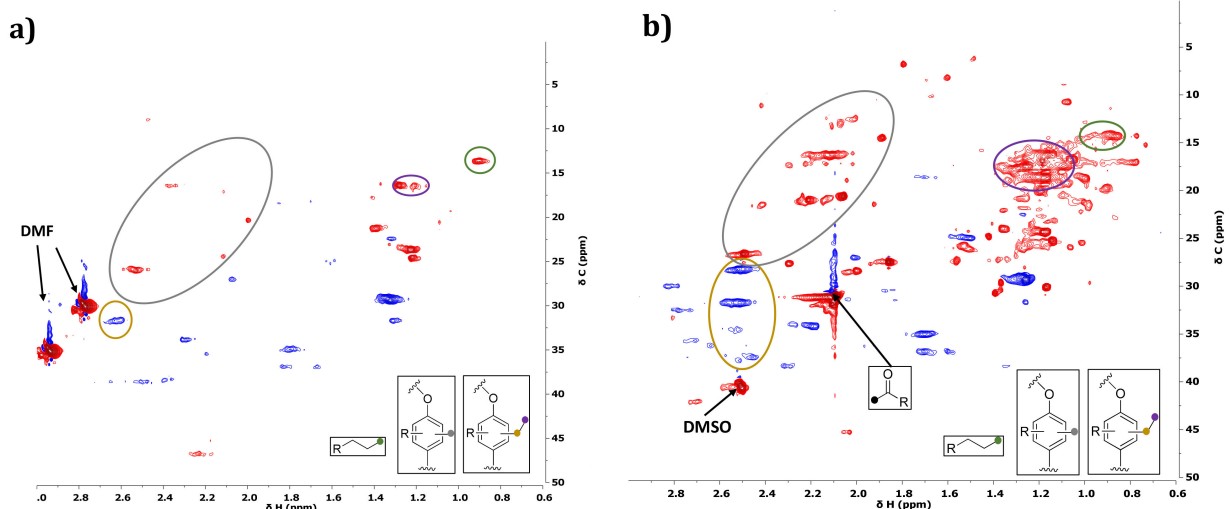

**Figure 16.** Side-chain region (non-oxygenated) of HSQC NMR spectra of initial (**a**) and organic phase after reaction without catalyst in water/EtOH (**b**).

It was also observed that the presence of alcohols produces higher amounts of ketones, the intense pic observed at 31/2.1 (δC/δH) can come from an alcohol reacting with itself or from other lignin derived ketones on the side-chain, like resulting from the

dehydrogenation of $C_{\alpha}HOH$ in the β-O-4 linkages. However, corresponding guaiacyl $G'_2$ (111.30/7.22 (δC/δH)) are not always observed. In all HSQC spectra, the guaiacyl units were in majority with a very small amount of hydroxyphenyl units (less than 5% of aromatic units).

The aliphatic oxygenated region shows also correlation for Aγ (59.1/3.26 and 3.30, δC/δH) in OP phase, however, with decreased intensity. Signal observed in OP phase, was rather attributed to propyl alcohol group on aromatics or as resulting from alcohol self-reaction.

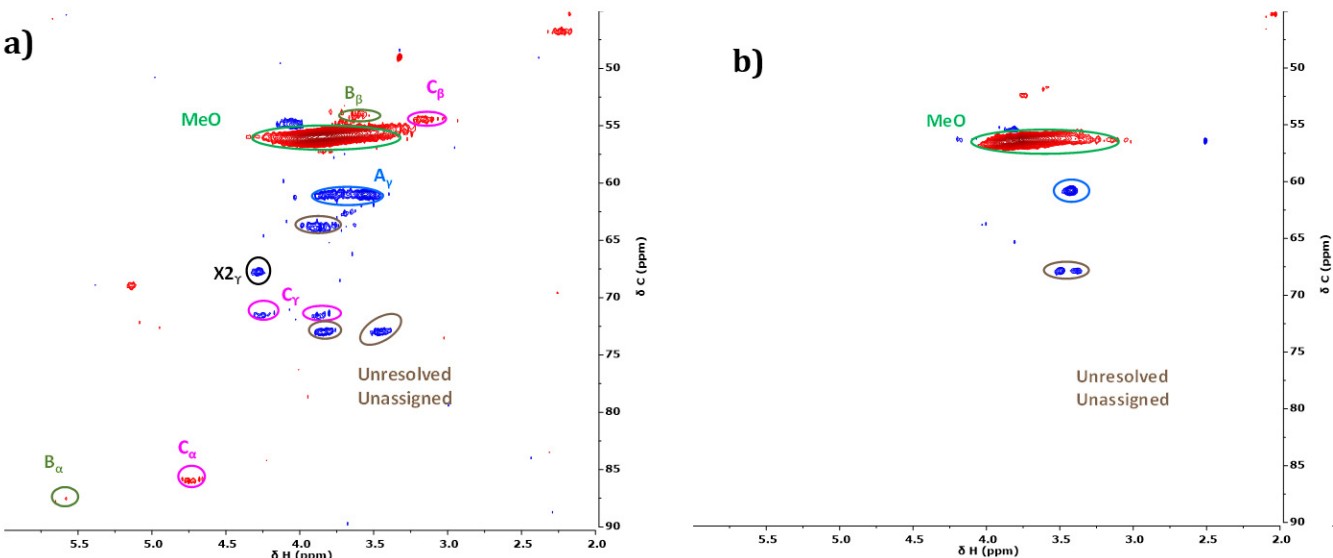

**Figure 17.** Side-chain region (oxygenated) of HSQC NMR spectra of initial (**a**) and organic phase after reaction without catalyst in water/EtOH (**b**).

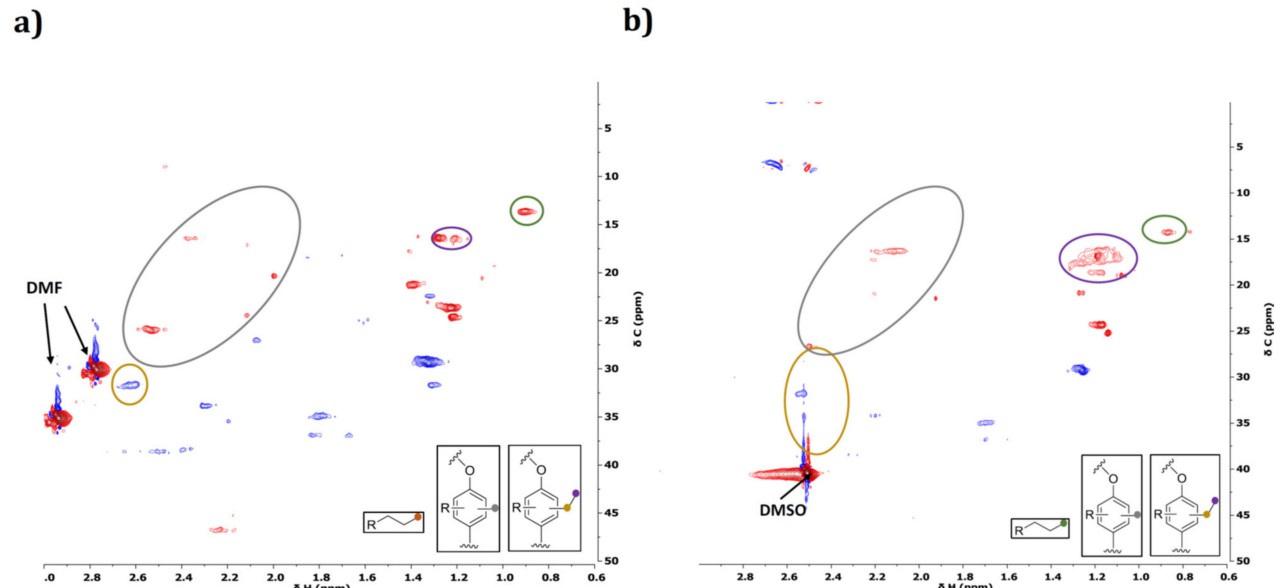

**Figure 18.** Side-chain region (non-oxygenated) of HSQC NMR spectra of initial (**a**) and Klason phase after reaction without catalyst in water/EtOH (**b**).

One can notice the disappearance of resinol and phenylcoumaran structures in the organic phases. Only in the organic phases from water/EtOH treatment, cinnamyl alcohol structures are preserved.

The correlations corresponding to the methoxy groups (56.58/3.72 ($\delta$C/$\delta$H)) remained intense for all the samples; hence, demethoxylation was very limited under our reaction conditions.

In summary, NMR analyses confirm, in accordance with GC/MS analyses, that that O-alkylation of the hydroxyl on the aromatic occurred. Probably rearrangements occurred during the reaction delivering the variety of detected alkylated compounds.

Detailed characterizations of reaction products revealed a limited oxidation of Klason phase, which remained similar to initial lignin. Organic phase is composed of oxidized and alkylated products, where $\beta$-O-4 linkages disappeared for the benefit of condensed C–C linkage.

### 2.3.4. Investigation of Pd/ZrO$_2$ Catalyst

Palladium was already reported as a catalyst for lignin conversion. For example, Samec showed the efficacy of palladium catalyst for the hydrogenolysis of $\beta$-*O*-4 linkages, one of the major linkages present in lignin [35]. Palladium nanoparticles and/or leached species were also reported to be active for ether bond cleavage of model lignin compounds [45]. While applicable to any supported metal used in aqueous solution, for palladium catalysts, the question that remains unanswered is whether the catalyst is performing in heterogeneous or homogeneous phase. When used in pure water, the amount of leached metals was measured in solution after treatment and was found to be 1.7% of total palladium amount in solution for Pd/ZrO$_2$, which is in the same range than for other evaluated catalysts, i.e., 2.8% for Pt/ZrO$_2$ and 4.4% for Ru/ZrO$_2$.

In order to gain insight on how truly the Pd/ZrO$_2$ catalyst is heterogeneous, the following reactions were performed under reference conditions (225 °C, 40 bar of argon for 3 h, the solution used is 50/50 water/EtOH with Kraft lignin):

- In the presence of ZrO$_2$ support alone, for comparison purpose (Run 1);
- In the presence of Pd/ZrO$_2$ and at the end of the reaction while the reactor is at 225 °C (hot filtration) sampling the reaction mixture (Run 2) for ICP analysis of the liquid phase;
- In the presence of the support ZrO$_2$ with the amount of palladium acetate (Pd(OAc)$_2$) corresponding to leached palladium in Run 2. Pd(OAc)$_2$ is entirely soluble in the reaction mixture at room temperature. This run was performed to determine the role of palladium and ZrO$_2$ during the catalytic test;
- In the presence of Pd(OAc)$_2$ in concentration corresponding to leached palladium in Run 2. Pd(OAc)$_2$ is entirely soluble in the reaction mixture at room temperature. This run was performed to determine the behavior of homogeneous palladium during the catalytic test (Table 6).

**Table 6.** Maximum amount of palladium in the reactor, on the support and in solution.

| Entry | Catalyst | Mass of Pd Introduced in Reactor | Pd in Solid Phase before Reaction | Pd in Solid Phase after Reaction |
|---|---|---|---|---|
| Run 1 | ZrO$_2$ | 0 | - | - |
| Run 2 | Pd/ZrO$_2$ | 50.1 ppm | 2.0%wt | 1.9%wt [a] |
| Run 3 | ZrO$_2$ + Pd(OAc)$_2$ | 0.54 ppm | 210 $\pm$ 5 ppm [b] | 220 $\pm$ 5 ppm |
| Run 4 | Pd(OAc)$_2$ | 0.50 ppm | n.d. | n.d. |

[a] Corrected for deposited carbon by TGA. [b] Calculated from the initial reaction mixture assuming that all introduced Pd as Pd(OAc)$_2$ would deposit onto the zirconia support.

Numerous reports showed that supported palladium catalysts can perform in a quasi-heterogeneous phase [46–48]. Indeed, the active phase can be in solution and redeposited on the surface of the support once the reaction is finished. One way of exploring the Pd dissolution–redeposition process is to sample the reaction mixture (filtered) while the solution is hot and quantifying the whole sample by ICP. Since some palladium particles might form during the cooling of the sample, the whole sample must be quantified. The

analysis of the solution in the presence of Pd/ZrO$_2$ showed the presence of palladium in solution around 4 ppm, which could be sufficient to catalyze homogeneous reactions, but most of the active metal remained on the support. When the reaction was performed in the presence palladium acetate (Pd(OAc)$_2$) and zirconia almost all palladium introduced was deposited on the support surface at the end of catalytic test. These results do not necessarily prove that the catalyst performs in homogeneous phase but clearly show that the leached palladium can be efficiently redeposited on the support inside the reactor.

The distribution of products of lignin solvolysis after Runs 1-2-3-4 was compared with the distribution of products in the absence of catalyst (Figure 19). The amount of organic phase was increased and the amount of aqueous phase was decreased in the presence of palladium, whatever its form. When the reaction was performed in the presence of ZrO$_2$, the results obtained were similar to the one without catalyst. The support itself does not seem to change dramatically the phase distribution. However, the presence of palladium decreases the yield in aqueous phase and this effect is more pronounced with Pd/ZrO$_2$. The formation of gaseous products or of char-like materials, enhanced by the presence of Pd/ZrO$_2$ catalyst, could explain this effect.

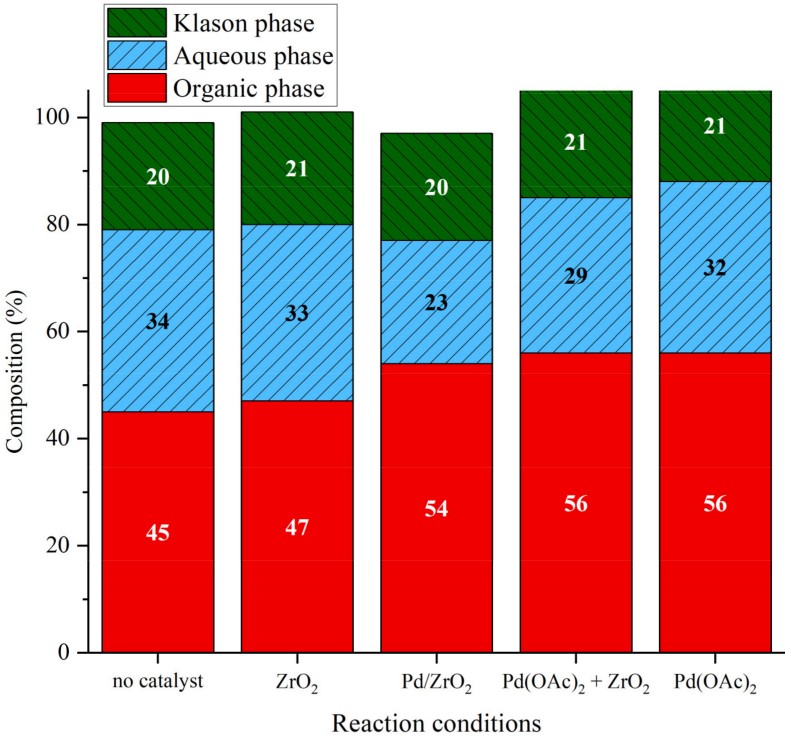

**Figure 19.** Phase compositions after lignin liquefaction in water/ethanol mixture over different catalytic systems. Reaction conditions: Kraft lignin 10 g·kg$^{-1}$, 3 h, 225 °C, 40 bar Ar.

The analysis of the organic phases (Figure 20) shows that depending on the catalyst used the monomers distribution shifts. When comparing the results obtained with Pd(OAc)$_2$ vs no catalyst, monomers yields are similar except for propylguaiacol (3.9 mg·g$_{lignin}$$^{-1}$ with Pd(OAc)$_2$ + ZrO$_2$, 3.6 mg·g$_{lignin}$$^{-1}$ with Pd(OAc)$_2$ alone and 2.8 mg·g$_{lignin}$$^{-1}$ without catalyst), showing a limited effect of homogeneous palladium except for propylguaiacol formation. When comparing the results obtained with Pd(OAc)$_2$ vs. Pd/ZrO$_2$ catalyst, a small decrease is observed for each monomer (except for alkyl esters of guaiacol, in traces amounts), with a total yield of 19 mg·g$_{lignin}$$^{-1}$ for Pd/ZrO$_2$ vs. 22 mg·g$_{lignin}$$^{-1}$ for Pd(OAc)$_2$. This most probably indicates the predominance of degradation and/or recondensation reactions catalyzed by Pd/ZrO$_2$. These degradation reactions could produce more gaseous or solid products, as observed above. When comparing the results obtained with Pd(OAc)$_2$ alone vs. ZrO$_2$ alone vs. Pd(OAc)$_2$ + ZrO$_2$, it is obvious that ZrO$_2$

alone or Pd(OAc)$_2$ alone do not favor the formation of monomers (ZrO$_2$ even decreases slightly the total yield in monomers), whereas the combination of metal and support favorably increases the amount of alkylguaiacols and esters of phenolic compounds (ethyl homovanilate and ethyl hydroferulate, cf ESI for detailed data).

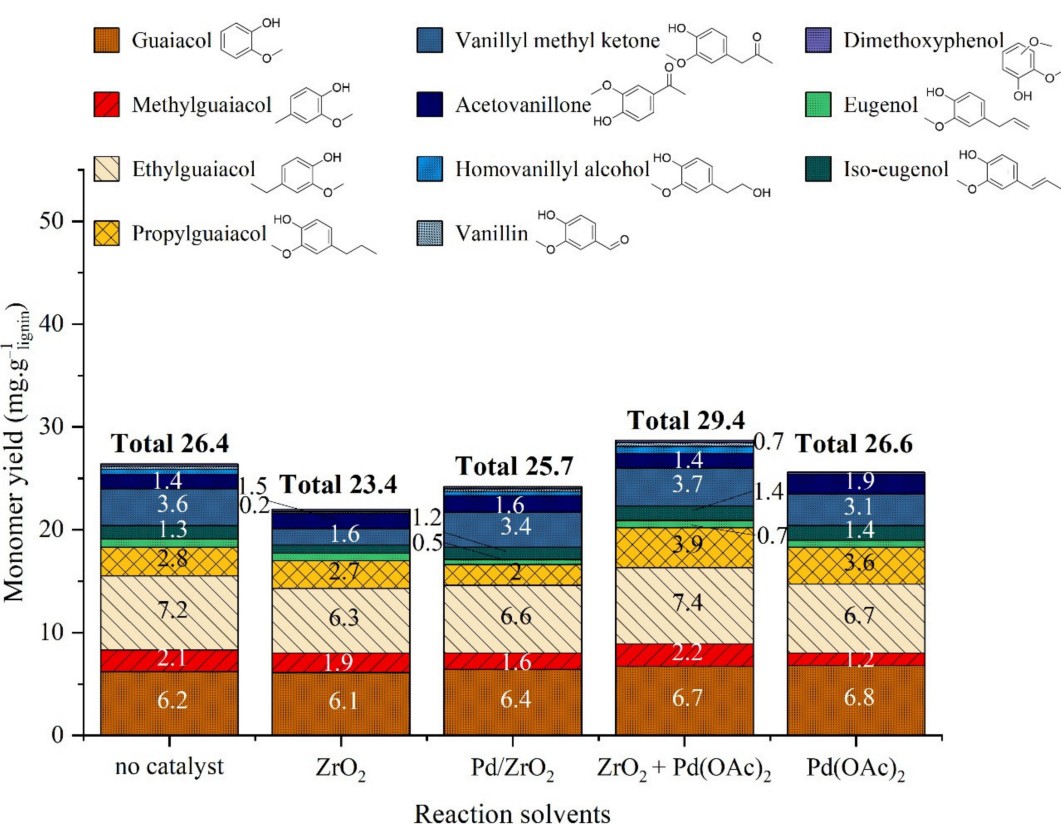

**Figure 20.** Yield of monomeric products in organic phase after catalytic lignin liquefaction in water/ethanol mixture over different catalytic systems. Reaction conditions: Kraft lignin 10 g·kg$^{-1}$, 3 h, 225 °C, 40 bar Ar.

The comparative study of palladium catalysts allows the following conclusions:

- Heterogeneous 2%Pd/ZrO$_2$ catalyst leads to global performances similar to homogeneous palladium 100 times less concentrated associated with ZrO$_2$. Thus, the depolymerization of lignin to organic and aqueous phases is suspected to be catalyzed by homogeneous Pd species, coming either from Pd/ZrO$_2$ leaching or Pd(OAc)$_2$ in solution. Homogeneous Pd or Pd nanoparticles in solution are supposed to diffuse more easily in tridimensional structure of lignin.
- The yield in monomers is higher in the absence of catalyst or in the presence of Pd(OAc)$_2$ in combination with ZrO$_2$. The degradation or recondensation of formed monomers could be catalyzed by Pd/ZrO$_2$ catalyst only, explaining the lower amount of monomers in this case.
- In the presence of H$_2$, (see Section 2.3.2) the yield in monomers increases with Pd/ZrO$_2$ with an evolution of products distribution. A higher rate of hydrogenolysis reactions catalyzed by Pd$^0$ heterogeneous species is assumed in this case.
- With the data collected, it is not possible to conclude on the type of active species when Pd(OAc)$_2$ is used as a source of palladium. At room temperature, Pd complexes form in solution; at the end of the reaction, most of the palladium is in solid phase, probably forming nanoparticles. The exact form of palladium in reaction conditions is unknown.

- The deposition of Pd on $ZrO_2$ during reaction could happen but does not influence the catalytic results, given that $Pd(OAc)_2$ leads to the same performances with and without $ZrO_2$.

## 3. Materials and Methods

### 3.1. General Information

All chemicals were used as received from suppliers (i.e., Alfa Aesar, Kandel, Germany; Carl Roth, Karlsruhe, Germany; Merck Chemicals, Fontenay sous Bois, France or Sigma-Aldrich, Saint Quentin Fallavier, France).

Lignin used in the study was purchased form Sigma-Aldrich as an Alkaline Kraft Lignin. It was found to be fully soluble in water up to solubility 270 g·$L^{-1}$.

### 3.2. Catalysts

All catalyst were prepared by precipitation–deposition procedure reported previously [49]. A suspension of the support was prepared in ultrapure water, an aqueous solution of the metallic salt ($K_2PtCl_4$, $RuCl_3$ and $PdCl_2$ the last two being solubilized using 10% HCl aqueous solution) was added under vigorous stirring to obtain approximately 2% metal loading. The suspension is stirred for 20 min and a solution of NaOH is added to reach a pH of 11. The solution is heated for an hour at 100 °C, the suspension is then cooled to ambient temperature. The sample is filtered then washed several times to remove the residual salts. The catalyst is then dried in a stove at 60 °C under nitrogen flow. The dry solid is reduced at 300 °C under hydrogen flow for 3 h and passivated using 1% $O_2/N_2$ flow.

### 3.3. Standard Procedure for Lignin Conversion

In a typical run, 1.5 g of alkaline lignin is dissolved either in 150 mL of deionized water, or alternatively in 75 g of deionized water and 75 g of alcohol (ethanol, methanol, isopropanol). At this time, the pH of the solution is around 9 without adding any base. The solution is introduced in the reactor and the catalyst $M/MO_x$ is added. The autoclave is purged three times with argon and pressurized to 40 bar argon. The reaction mixture is stirred and heated to 225 °C and held under these conditions for the desired time. Once the reaction is finished, the autoclave is cooled to room temperature with ice bath. The liquid phase is recovered and the reactor is rinsed with 50/50%wt water-ethanol solution, and a basic solution (0.1% NaOH) is introduced and heated at 150 °C for 4 h for cleaning it before next run.

### 3.4. Products Recovery

The protocol for products recovery is summarized in Figure 21. An aliquot of 50 mL of the solution was collected from reaction medium, it was filtrated to remove the catalyst and then precipitated with 10% HCl solution at pH 1, and 10 mL of water was added to facilitate precipitation. The solution was shacked for 2 min, and a color change from dark brown to light brown was observed at this stage, which is an indicator of the lignin precipitation. After centrifugation at 4000× *g* rpm for 10 min, the solid obtained was washed with water, dried under vacuum and weighed to estimate the precipitable matter (Klason phase). The liquid phase was first extracted with three times 50 mL dichloromethane. Collected fractions were concentrated on a rotary evaporator and dried under reduced pressure giving Organic Phase fraction. After weighing, 2 mL of acetonitrile was added to the Organic Phase, together with the internal standard (toluene), before being analyzed with GC-FID/MS. The aqueous phase contained fragments of lignin soluble in water and aliphatic polar compounds; when applicable some alcohol degradation products, and residual salts from the Kraft process and Cl from the hydrochloric acid. After evaporation of water, the residue was taken with THF (tetrahydrofuran) to remove salts. THF was then evaporated delivering the Aqueous phase fraction.

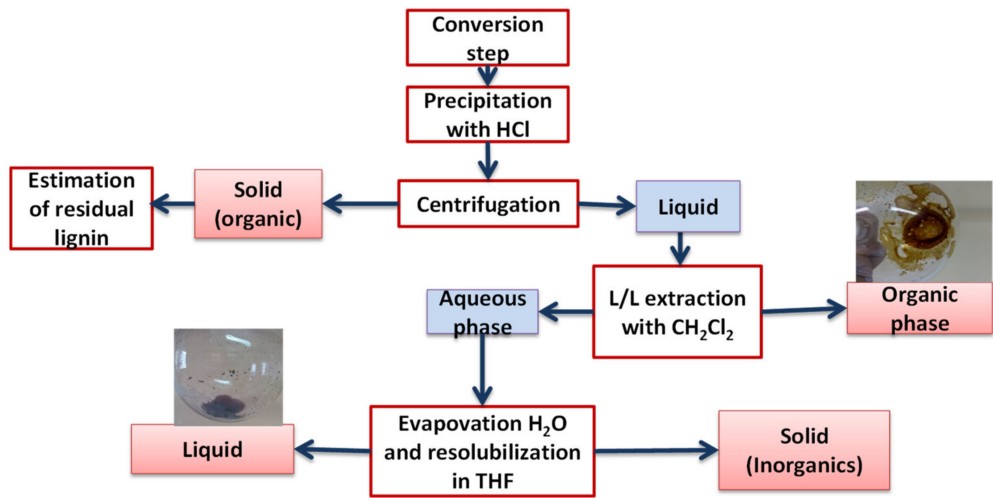

**Figure 21.** Protocol for products recovery.

The accuracy of the fractionation and the analytical tools were determined by performing three runs under the same conditions. The sum of the different phases gave a mass balance of at least 80%, several factors could be attributed for this loss: insoluble matter formed on the reactor wall and the stirrer, formation of volatile components lost during the in-vacuum drying of the different fractions and the production of gas.

*3.5. Analytical Methods*

Product quantification was conducted using a Shimadzu 2010 GC-FID (Shimadzu Europa GmbH, Duisburg, Nordrhein-Westfalen, Germany) equipped with a Phenomenex Zebron ZB-5HT capillary column (5% Phenyl-95%dimethylsiloxane, 30 m × 0.25 mm × 0.25 μm) (Phenomenex France, Le Pecq, France), and for the product identification a Shimadzu GC-MS QP2010 (Shimadzu Europa GmbH, Duisburg, Nordrhein-Westfalen, Germany) equipped with Supelco SLB-5MS column (5% Phenyl-95%dimethylsiloxane, 30 m × 0.25 mm × 0.25 μm) was used. Helium was used as carrier gas with the following program: 60 °C for 2 min/Heat up to 150 °C (rate 30 °C/min)/Heat up to 200 °C (rate 2 °C/min)/Heat up to 300 °C (rate 10 °C/min)/300 °C for 2 min). Samples were dissolved in acetonitrile before analysis. Product identification was possible by comparing MS data with the NIST 2017 Mass database. Toluene was used as an internal standard.

FTIR spectroscopy measurements were performed using a Vector 22 spectrometer (Bruker Corporation, Billerica, MA, USA), 100 scans from 400–4000 cm$^{-1}$ were collected. Pellets were prepared by mixing 2 mg of lignin sample with 400 mg of KBr and an aliquot of 200 mg used to prepare one pellet.

NMR analyses were performed with a Bruker AVANCE III 400 MHz spectrometer (Bruker Corporation, Billerica, MA, USA) equipped with a BBFO probe (Z gradient). HSQC NMR analyses acquisitions were realized at 50 °C (48 scans). Lignin samples (ca. 50 mg) were dissolved in 600 mg of DMSO-d$^6$. Selected samples were analyzed with the $^{31}$P NMR technique to quantify the OH groups of PM and OP fraction (150 scans) [50–53]. 2-chloro-4,4,5,5-tetramethyl-1,3,2-dioxaphospholane (TMDP) is used as phosphytilation agent to quantify the various OH groups (phenol, aliphatic, carboxylic). Samples were accurately weighted (ca. 30 mg) and solubilized in 0.5 mL of a solution of cyclohexanol (3.95 mg/g$_{solution}$—internal standard) in pyridine and DMF (*v/v*: 1/1). Once dissolved, 200 mg of CDCl$_3$ were added before introducing 100 mg of TMDP. The product of the reaction of TMDP with water (δ = 132.2 ppm) was used to calibrate the chemical shifts.

The metal loading of catalysts was determined by inductively coupled plasma optical emission spectrometer (ICP-OES, Activa Jobin-Yvon) ( HORIBA Scientific, Longjumeau, France) after mineralization following a specific procedure for the metal. For platinum and palladium-based catalysts, the samples are digested in mixture of acids ($H_2SO_4$ + HCl + $HNO_3$) and

heated at 250–300 °C. For ruthenium catalysts, the samples are digested in bomb with a mixture of $H_2SO_4$ and $HNO_3$ and heated at 150 °C.

Specific surface areas were calculated following the BET procedure. Measurements were done using $N_2$ adsorption–desorption at −196 °C on a Micromeritics ASAP 2020 (Micromeritics France SARL, Mérignac, France). Before analysis the catalyst samples were degassed at 150 °C under vacuum ($10^{-3}$ Torr) for 3 h. Pore size distribution was obtained by using the BJH pore analysis applied to the desorption branch of the nitrogen adsorption/desorption isotherm.

## 4. Conclusions

In this study, we were able to see the effect of different systems of water and alcohol as a co-solvent for the conversion of a Kraft lignin under sub-critical conditions (225 °C, 40 bar) in a batch reactor. In hydrothermal conditions (water only), the solubilization of Klason lignin occurred but the conversion of lignin to monomers remained limited. Vanillin was the major monomer formed. Adding a metal supported catalyst led to a sharp increase in monomers yield, with guaiacol as major identified monomer. $ZrO_2$ was the best support for monomers production, enhancing catalytic activity better than $Al_2O_3$ or $TiO_2$. Palladium exhibited superior catalytic activity when compared to platinum and ruthenium. $Pd/ZrO_2$ was the best compromise between catalytic production of guaiacol and vanillin and limited degradation of monomers such as vanillin. It produced 15.5 mg·$g_{lignin}^{-1}$ of monomers. Therefore, metal catalysts seem to modify the reaction pathway for monomer production: oxidation products are formed by non-catalytic reactions whereas hydrogenolysis products are formed through catalytic reactions.

When an alcohol was added as a co-solvent, a remarkable increase in organic phase (oligomers) production was observed. The nature of co-solvent had a strong impact on lignin solvolysis, with EtOH giving the highest Klason lignin conversion and organic phase production. Alcohols also enhanced the production of monomers: guaiacol and alklyl-guaiacols were produced in high amounts, particularly when iso-propanol was used as a co-solvent, reaching 48 mg·$g_{lignin}^{-1}$ at 225 °C. Adding a $Pd/ZrO_2$ catalyst in reaction medium during lignin solvolysis in alcohol/water mixture resulted in a higher depolymerization of lignin to organic phase (oligomers) products but did not dramatically improve the production of monomers, whatever the solvent.

Detailed characterization of reaction products revealed a partial oxidation Klason phase and Organic phase. Alkylation of phenolic moieties in Organic phase also occurred when alcohol was used as a co-solvent. The final oligomers and monomers did not contain any β-O-4 linkages.

The role of palladium catalyst was investigated using homogeneous or heterogeneous palladium species. We conclude that primary depolymerization of lignin occurs mainly with homogeneous palladium species. Degradation of monomers occurs mainly on heterogeneous $Pd/ZrO_2$. $ZrO_2$ also played a role in the formation of monomers. The highest yield of oligomers was obtained with $Pd(OAc)_2$ as a catalyst and the highest yield of monomers was obtained with a system $Pd(OAc)_2$ + $ZrO_2$.

Finally, the results obtained highlighted the importance of the following operation conditions during lignin catalytic solvolysis: type of co-solvent, temperature of reaction and catalytic system.

**Supplementary Materials:** The following are available online at https://www.mdpi.com/article/10.3390/catal11080875/s1, File "ESI". This work comes from Woldemichael Sebhat's PhD thesis (Université Claude Bernard Lyon 1, France). The entire dissertation can be found here: https://tel.archives-ouvertes.fr/tel-01297040 (accessed on 15 July 2021)

**Author Contributions:** Conceptualization, W.S., L.D. and P.F.; methodology, W.S., L.D. and P.F.; investigation, W.S. and A.E.R.; data curation, W.S., L.D. and L.V.; writing—original draft preparation, W.S.; writing—review and editing, L.V. and L.D.; visualization, W.S., L.D. and L.V.; supervision, P.F.

and L.D.; project administration, L.D. and P.F.; funding acquisition, L.D. and P.F. All authors have read and agreed to the published version of the manuscript.

**Funding:** The authors gratefully acknowledge the French National Agency of Research (CHEMLIVAL N° ANR-12-CDII-0001_01) for funding. W.S. thanks the French Ministry of Research and Education for PhD grant.

**Acknowledgments:** All the authors thank the analytical services of IRCELYON for their help.

**Conflicts of Interest:** The authors declare no conflict of interest. The funders had no role in the design of the study; in the collection, analyses, or interpretation of data; in the writing of the manuscript, or in the decision to publish the results.

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
