# Peer review of "Catalytic Liquefaction of Kraft Lignin with Solvothermal Approach"

_catalysts, doi:10.3390/catal11080875_

Round 1
Reviewer 1 Report
Identification of some efficient and sustainable methods for decomposing recalcitrant lignin and hemicelluloses structures is a research topic of major importance in the recovery of bioresources for the production of various chemicals, as well as in the pretreatment stages of lignocellulosic biomass conversion into biofuels. The topic addressed by the authors is particularly useful for the green chemicals and energy sectors.
Introduction is properly structured and elaborated with professionalism; Reference is made to the chemical methods currently used for the depolymerization of lignin under different processing conditions, but the main limitations regarding the decomposition of lignocellulose are also briefly mentioned. The aim of the paper is clearly defined.
The materials used in the experimental work are presented properly, the research methodology is described in detail and accurately in section 3. Also, the authors present the analytical methods used for the chemical products characterization. The information provided in this section are structured fluently and coherently, allowing the easy understanding of this complex research. However, in my opinion this section would be more appropriate to be placed before chapter 2 regarding results and discussions.
The experimental part of this study is quite complex and carried out in a logical sequence of operations, the results regarding the solubilization of lignin in various decomposition conditions are analyzed with professionalism. The very good quality of the images and the graphic representation of the measured parameters are able to support the scientific comments offered by the authors.
The practical importance of this research is obvious, especially in the context of joint efforts to identify new raw materials that are widely available and non-polluting. Apart from the remark regarding the placement of sections 2-3, I have no suggestions for improving the manuscript. It is a scientific work developed and presented highly satisfactorily.
Author Response
We thank Reviewer 1 for his/her kind appreciation of our work. Regarding the order of sections in the manuscript, we followed the Template of Catalysts journal with Results section placed before Experimental section.
Reviewer 2 Report
In this research article entitled “Catalytic Liquefaction of Kraft Lignin with Solvothermal Approach”, Laurent Djakovitch and co-workers reported a very interesting study on the solvolysis of Kraft lignin in water and water/alcohol mixtures for the production of oligomers and monomers of interest. In particular, the impact of the presence of several supported palladium catalysts was evaluated: the yield of monomers was particularly enhanced in the presence of Pd/ZrO2 catalyst.
In my opinion, the research design of the work is appropriate and the results are clearly presented: the paper is certainly interesting for scientists expert in the field and it perfectly fits with the scope of the journal. Therefore, I have no doubts in recommending its publication in the MDPI Catalysts journal, after the following very few minor revisions.
1) In the introduction, I suggest to add some more discussion on the role of the supported palladium catalyst in lignin solvolysis. In this context, I think that some other representative references on the topic could also be adde (for example, I suggest: DOI: 10.3390/cleantechnol2040032 concerning a complete overview of all the catalytic systems used in the solvolysis of Kraft lignin; DOI 10.1002/slct.201601736 concerning palladium supported on both inorganic and organic matrices; DOI: 10.3390/catal9010043 concerning catalytic transformations of lignin to fuels and chemicals).
2) Other minor issues. (i) In Table 2, some of the names of monomeric products are in capital letter, while others are in lowercase letter; I think that they could be homogenized. (ii) In Table 6, caption a, please change “corrected” with “Corrected”. (iii) In Figure 20, please correct Pd(OAc)2 with Pd(OAc)2.
Author Response
We thank Reviewer 2 for his/her kind evaluation of our work. Regarding the specific comments:
1) In the introduction, references suggested by Reviewer 2 were added line 80, page 3 (refs 20 and 22). The last reference was added in the part corresponding to palladium studies (p. 22, l.585): "Palladium nanoparticles and/or leached species were also reported to be active for ether bond cleavage of model lignin compounds [45]."
2) (i) In table 2, all compounds were written with a first capital letter (Table 2, p.7)
(ii) in Table 6 captions, "corrected" was replaced by "Corrected" (Table 6, p.23)
(iii) in Figure 20, Pd(OAc)2 was replaced by Pd(OAc)2 (Fig 20, p.25 and separated file)
Reviewer 3 Report
In this paper, the authors performed the solvolysis of Kraft lignin ysing water and in water/alcohol mixtures to produce oligomers and monomers of interest, at mild temperatures (200-275°C) under inert 14 atmosphere. I agree that this topic is very relevant for the catalysis and and environmental fields. This work is well written and presented, and the authors provided very amount of experimental data when compared to other previous works reported in literature. Minor revision is need before I recommend its publication.
Specific comments 1 – In introduction, clearly build your research hypothesis (straightforward question that is answerable by yes or no). I am not sure this is clear in the manuscript. 2 – Please underscore the novelty of your work in the introduction part. In the introduction, explain the main differences between your work and the ones found in literature 3– The other consideration is about to make a comparison, in a table, highlighting the main findings in the literature with the findings of this manuscript. The idea is to compare the effciency of the catalytic processes with others found in the literature.
4 – Please cite at least 5 recent papers (2020-2021) from Catalysts.
Author Response
We thank Reviewer 3 for his/her kind evaluation of our work. Regarding specific comments:
1- Our research goal is given in l.103-105 of the manuscript: "The aim of this work is the investigation of Kraft lignin solvolysis in water and organic solvents in the absence or presence of metal supported catalysts. The roles of solvent, metal and supports in lignin solubilization and depolymerization are detailed."
2- The novelty of our work was detailed l.99-102, p. 3, in the introduction: "Despite numerous works on heterogeneous catalysis for lignin solvolysis, there is a lack of comparative studies of metals and oxide supports used as catalysts for technical lignin solvothermal liquefaction."
3- It is very difficult to make an accurate comparison between lignin valorisation works because "lignin" is not a standard material and can recover various types of biopolymers with various features, depending on the nature of intial biomass, conditions of extraction, conditions of purification, etc. Some remarkable reviews address this point (for example Schutyser et al, Chem Soc Rev 2018).
4- References were added following Reviewer 2 recommendations. If Reviewer 3 wants to make specific recommendations for citations, we would be glad to add them to our manuscript. However, I believe that articles should not be cited based only on the Journal they were published in.